# El Niño–Southern Oscillation (*ENSO*) event reduces CO₂ uptake of an Indonesian oil palm plantation

Christian Stiegler[1], Ana Meijide[2], Yuanchao Fan[3], Ashehad Ashween Ali[1], Tania June[4], Alexander Knohl[1]

[1]Bioclimatology, University of Goettingen, Goettingen, Germany
[2]Department of Crop Sciences, Division Agronomy, University of Goettingen, Germany
[3]NORCE Norwegian Research Centre, Bjerknes Centre for Climate Research, Bergen, Norway
[4]Department of Geophysics and Meteorology, Bogor Agricultural University, Bogor, Indonesia

*Correspondence to*: Christian Stiegler (christian.stiegler@biologie.uni-goettingen.de)

**Abstract.** The El Niño–Southern Oscillation (*ENSO*) in 2015 was one of the strongest observed in almost 20 years and set the stage for a severe drought and the emergence of widespread fires and related smoke emission over large parts of Southeast Asia. In the tropical lowlands of Sumatra, which were heavily affected by the drought and haze, large areas of tropical rainforest have been converted into oil palm (*Elaeis guineensis* Jacq.) plantations during the past decades. In this study, we investigate the impact of drought and smoke haze on the net ecosystem $CO_2$ exchange, evapotranspiration, yield and surface energy budget in a commercial oil palm plantation in Jambi province (Sumatra, Indonesia) by using micrometeorological measurements, the eddy covariance method, yield data and a multiple linear regression model (*MLRM*). With the *MLRM* we identify the contribution of meteorological and environmental parameters to the net ecosystem $CO_2$ exchange. During the initial part of the drought, when incoming shortwave radiation was elevated, net $CO_2$ uptake increased by 50% despite a decrease in upper-layer soil moisture by 35%, an increase in air temperature by 10% and a tripling of atmospheric vapour pressure deficit. Emerging smoke haze decreased incoming solar radiation by 35% compared to non-drought conditions and diffuse radiation became almost the sole shortwave radiation flux for two months resulting in a strong decrease in net $CO_2$ uptake by 86%. Haze conditions resulted in a complete pause of oil palm net carbon accumulation for about 1.5 months and contributed to a decline in oil palm yield by 35%. With respect to a projected pronounced drying trend over the western Pacific during future El Niño, our model showed that an increase in drought may stimulate net $CO_2$ uptake while more severe smoke haze, in combination with drought, can lead to pronounced losses in productivity and net $CO_2$ uptake, highlighting the importance of fire prevention.

## 1 Introduction

El Niño – Southern Oscillation (*ENSO*) is a coupled ocean-atmosphere interaction phenomenon in the equatorial Pacific Ocean and one of the most distinct drivers of seasonal to interannual regional and global climate variability (Wolter & Timlin, 2011). Increasing sea surface temperatures in the eastern and central tropical Pacific Ocean are linked to increases in sea-level air pressure in the western Pacific Ocean resulting in reduced cloudiness and low precipitation over Southeast Asia (Rasmusson

& Carpenter, 1981; Wolter, 1986). Generally, *ENSO* shows episodic and varying timing, frequencies and amplitudes but *ENSO* during 2015 was the strongest observed in almost 20 years (Santoso et al., 2017; Lim et al., 2017). It set the stage for a severe drought over large parts of Southeast Asia, particularly in Indonesia, which favoured the emergence of widespread and mostly human-induced forest, grassland and peat fires (Betts et al., 2016).

The fires released record-breaking amounts of terrestrial-stored carbon as $CO_2$ into the atmosphere, with mean daily emission rate of 11.3 Tg $CO_2$ during September to October 2015 (Huijnen et al., 2016). The recent *ENSO* elevated Mauna Loa mean monthly $CO_2$ concentration for 2015 above 400 ppm for the first time in its measurement history and contributed to the highest annual $CO_2$ growth rate on record (Betts et al., 2016). The emitted aerosol particles from biomass burning covered large parts of Sumatra, Borneo, Malay Peninsula and Singapore for several months under a persistent pall of smoke haze.

The regions affected by the smoke haze, especially Indonesia and Malaysia, have undergone substantial land-use changes within the past two decades due to the world's hunger for cheap vegetable oil, such as palm oil (Koh et al., 2011). Oil palm (*Elaeis guineensis* Jacq.) emerged to an important cash crop due to the extensive application of palm oil in pharmaceutical, cosmetics and food industries as well as for biofuel (Koh & Ghazoul, 2008; Turner et al., 2018). Indonesia and Malaysia are the world's biggest producers of palm oil. For example, in 2016/17, the two countries contributed 56% (Indonesia) and 30%

(Malaysia) to the global supply of palm oil (USDA, 2018). In 2015, oil palm plantations in the two countries combined covered 17 Million hectares (Chong et al., 2017).

Oil palm has high life cycle of about 25 years (Woittiez et al., 2017) and is adapted to tropical climate with optimal mean temperature of 24-28°C, it requires frequent and sufficient precipitation of ~2000 mm yr$^{-1}$ and high level of solar radiation (Bakoumé et al., 2013; Corley & Tinker, 2016). Oil palm shows distinct reaction to changes in atmospheric and soil parameters,

with gradual symptoms of water and heat stress such as inhibited growth (Legros et al., 2009; Cao et al., 2011), snapping off leaves and drying out of fruit bunches (Bakoumé et al., 2013), reduction in yield (Caliman & Southworth, 1998; Noor et al., 2011), reduction or even pause in carbon dioxide assimilation (Méndez et al., 2012; Jazayeri et al., 2015) and ultimately, plant death (Maillard et al., 1974).

Aerosol particles from biomass burning generally reduce the amount of sunlight reaching the surface and increase the fraction

of diffuse radiation through scattering (Kozlov et al., 2014). Diffuse light conditions up to a certain level enhance plant photosynthesis and evapotranspiration through more uniform through-canopy distribution of photosynthetically active radiation (*PAR*) (Knohl & Baldocchi, 2008; Kanniah et al., 2012; Heuvelink et al., 2014). Light haze smoke intensities may therefore increase $CO_2$ uptake, maximum rate of photosynthesis ($A_{max}$) and evapotranspiration but during dense haze smoke, the effect is reversed due to the overall reduction of incoming *PAR* (Yamasoe et al., 2006; Moreira et al., 2017). In addition,

ambient atmospheric $CO_2$ increase due to local fires and burning may act as a temporary plant $CO_2$ fertilization which, to some extent, may offset reduced plant $CO_2$ uptake during dense smoke haze (Mathews & Ardiyanto, 2016).

Global warming and consequent regional climate changes, including changes in precipitation pattern and increase in the magnitude and frequency of extreme events such as drought, *ENSO* and fires (Neelin et al., 2006; IPCC, 2013; Jiménez-Muñoz et al., 2016), may severely stress oil palm plantations in the near future (Tangang, 2010; Rowland et al., 2015). It is therefore

important to assess how much net ecosystem $CO_2$ exchange (*NEE*) would change under such conditions. Model predictions suggest more intense *ENSO* over the course of the 21[st] century, which may result in a general drying in the western regions of the Pacific Ocean during El Niño (Power et al., 2013; Cai et al., 2014; Kim et al., 2014; Keupp et al., 2017; Cai et al., 2018). Increasing frequency of *ENSO*-related drought in Southeast Asia has already caused a decline of 10-30% in palm oil production (Paterson et al., 2017). Projected temperature increase and water stress through enhanced *ENSO* might further decrease oil palm yield (Oettli et al., 2018) or even lead to detrimental conditions for oil palm growth in some areas in Southeast Asia (Paterson et al., 2017). On the other hand, *ENSO* is associated in Indonesia with an increase in incoming solar radiation which can increase $CO_2$ uptake in a tropical environment (Olchev et al., 2015). However, current studies and modelling approaches lack a holistic understanding of ecosystem response, resilience and the underlying meteorological, ecological and biological processes during extreme events, such as drought and smoke haze conditions. The *ENSO* in 2015 was the first strong climate extreme event after the major land-use conversions on Sumatra from forest into oil palm plantations but only little is known about how the *ENSO*-related severe drought and persistent smoke haze influenced oil palm monoculture.

In this study, we therefore aim to (a) quantify land-atmosphere $CO_2$, water vapour and turbulent heat exchange over oil palm plantation using the eddy covariance technique during the 2015-*ENSO*, (b) analyse the contribution to net ecosystem $CO_2$ exchange (*NEE*) of meteorological and environmental parameters using a multiple linear regression model (*MLRM*), (c) investigate the impact of a possible near-future more severe drought and smoke haze scenario on *NEE* and (d) evaluate potential changes in evapotranspiration and energy fluxes to the atmosphere. We hypothesize that (a) oil palm monoculture would reduce net ecosystem $CO_2$ uptake and maximum photosynthetic rate ($A_{max}$) during drought and haze, and (b) sensible heat fluxes would increase at the cost of evaporative cooling.

## 2 Materials and methods

### 2.1 Study site

The study site is located in a commercial oil palm plantation (1°41'35.0"S, 103°23'29.0"E, 76 m a.s.l.) in tropical lowlands of Jambi province on Sumatra island (Indonesia), approx. 25 km west-southwest of Jambi City (Figure 1). The landscape is flat with small elevation variations of approx. ± 15 m. Average mean annual air temperature during the period 1991-2011 is 26.7°C (± 0.2°C standard deviation) and mean precipitation for the same period is 2235 mm yr$^{-1}$ (± 381 mm SD), with a dry season from June to September and two peak rainy seasons around March and December (Drescher et al., 2016). Long-term climate records are collected at Sultan Thaha Airport Jambi, approx. 29 km east-northeast of the study site. A comparison of air temperature and precipitation at our study site with climate records from Sultan Thaha Airport Jambi during our study period May 2014 to July 2016 showed no significant differences in daily average air temperature (P<0.001) or in monthly accumulated precipitation (P<0.001). Therefore, we consider the long-term climate records being representative for our study location. The oil palm plantation covers 2186 ha and the palm seedlings were planted in the years 1999, 2002 and 2004. Our measurements are located in the section where the palms have been planted in 2002. Palms are planted in a triangular array,

with 8×8 m horizontal density and 156 palms per ha. Based on this horizontal density, an average palm height of 12 m, and 35-45 expanded leaves per palm, Fan et al. (2015) estimated a site-specific leaf area index (*LAI*) of 3.64 $m^2$ $m^{-2}$. Gaps in oil palms that can be created due to disturbances or extreme weather conditions were not observed in this study. In 2015, 144 kg $ha^{-1}$ of Magnesium Nitrate, 575 kg $ha^{-1}$ of NPK Granular, and 251 kg $ha^{-1}$ of Dolomite fertilizers were applied in topdress

application. The plantation is owned by Perseroan Terbatas Perkebunan Nusantara VI, Batang Hari Unit (PTPN6). Stumps of pruned oil palm leaves are densely covered with epiphytes, e.g. ferns (*Polypodiophyta*) or flowering plants (*Melastomataceae*, *Orchidaceae*), while understory vegetation is scarce due to regular application of herbicides and occasional mowing. Highly weathered Loam Acrisols soils dominate in the area (Allen et al., 2015) and mean soil carbon and nitrogen content in the plantation reach 1.12% (± 0.34% SD) and 0.08% (± 0.02% SD) (Meijide et al., 2017).

**2.2 Eddy covariance measurements**

Eddy covariance (*EC*) measurements to derive fluxes of sensible (*H*) and latent (*LE*) heat, net ecosystem $CO_2$ exchange (*NEE*) and water vapour (*ET*) for this study were carried out from June 2014 to July 2016. We use a LI7500A fast response open-path $CO_2/H_2O$ infrared gas analyser (LI-COR Inc. Lincoln, USA) and a Metek uSonic-3 Scientific sonic anemometer (Metek, Elmshorn, Germany). The *EC* system measures at 10 Hz and is placed at the top of a 22 m high steel framework tower. Digital

signal recording, statistical tests for raw data screening and raw data correction, spectral analysis, eddy flux calculation using EddyPro (LI-COR Inc, Lincoln, USA), post-processing such as quality flagging, removal of fluxes during stable atmospheric conditions, i.e. friction velocity (*u\**) <0.1 m $s^{-1}$, flux footprint analysis and gap filling of missing flux data follow standard procedures (Meijide et al., 2017). The energy balance closure for the entire study period was 0.75 ($R^2$ = 0.85).

**2.3 Meteorological ad environmental parameters, oil palm yield**

Above-ground measurements include air pressure (22 m above the surface), precipitation (11.5 m), wind direction (15.4 m) and wind speed (18.5, 15.4, 13 and 2.3 m), air temperature and air humidity (22, 16.3, 12.3, 8.1, 2.3 and 0.9 m), incoming and reflected photosynthetically active radiation (*PAR*) (22 m), incoming and outgoing shortwave and longwave radiation (22 m), global and diffuse radiation (22 m), and sunshine duration (22 m). Detailed information on instrument type and manufacturer for all measured parameters can be found in Meijide et al., (2017). Below-ground measurements consist of three profiles where

ground heat flux (*G*) is measured with heat flux plates at 5 cm depth and soil moisture and soil temperature is measured at 0.3, 0.6 and 1 m depth, respectively. All meteorological and environmental parameters were measured every 15 s and stored as 10-minute mean, minimum and maximum values in a DL16 Pro data logger (Thies Clima, Göttingen, Germany). Monthly oil palm yield data was provided by PTPN6 and covers the period January 2013 to April 2017.

**2.4 Data analysis and statistics**

The meteorological data used in this study covers the period from May 2014 to July 2016. Based on precipitation and the ratio between diffuse and global radiation ($R_G$), i.e. fraction of diffuse radiation (*fdifRad*), we defined four distinct meteorological

periods during 2015, i.e. pre-drought, non-haze drought, haze drought, and post-haze and compared the four periods with meteorological conditions in 2014 and 2016. We consider pre-drought as the period with frequent precipitation on an almost daily basis and non-haze drought as the period when precipitation occurred only sporadically and heavy precipitation events >50 mm d$^{-1}$ were completely absent. Haze drought period follows the non-haze drought. We defined the start of the haze drought period at the day when daily average fraction of diffuse radiation was >0.8 for more than three consecutive days. We consider the end of the haze drought period as the day when daily average fraction of diffuse radiation dropped below 0.8 for five consecutive days and when clear day-to-day variations in fraction of diffuse radiation, with day-to-day variation of >0.2 became apparent. Reference meteorological conditions cover the period May-December 2014 and January-July 2016.

To investigate the behaviour of the oil palm plantation in more detail, we defined day (6-18:30 h local time), night (19-5:30 h) and midday (10-14 h) time periods. Due to the proximity of our study site to the equator the difference in day length between summer and winter solstice is only 12 minutes. Therefore, we consider the impact of differences in day length on the fluxes and meteorological parameters as negligible.

Maximum rate of photosynthesis ($A_{max}$) at ecosystem scale was calculated from daily light response curve using $NEE$ (Falge et al., 2001). Initially, we applied $CO_2$ flux partitioning of $NEE$ into gross primary production ($GPP$) and respiration using (a) non-linear regression model based on Reichstein et al. (2005) and (b) $CO_2$ flux partitioning based on $CLM$-$Palm$ (Fan et al., 2015) which is a sub-model within the framework of the Community Land Model ($CLM4.5$) (Oleson et al., 2013). The non-linear regression model underestimated $NEE$ by 58%, on average, most likely because the model struggles to assess the temperature sensitivity of ecosystem respiration using the filtered nighttime data (Oikawa et al., 2017). $CLM$-$Palm$ struggled to represent daily average $NEE$ during the non-haze drought and haze drought periods, most likely due to the models' soil water stress function (Sellers et al., 2013) and missing plant hydraulic processes in the overarching $CLM4.5$ (Oleson et al., 2013). Therefore, we decided to solely focus on $NEE$ to describe the overall $CO_2$ flux behaviour of the oil palm plantation during the extreme events of drought and haze. However, we used the nighttime $NEE$ (=respiration) as a proxy for the overall behaviour of oil palm monoculture respiration and disentangled its driving climatic variables. Seasonal differences in $u^*$, especially during nighttime, might impact the performance of eddy covariance gap filling. However, we found no significant differences ($P<0.05$) in $u^*$ which could have affected the proportion of available nighttime data during the different meteorological periods. Therefore, we consider the applied gap filling procedure and derived flux averaging as robust and representative for the studied time periods.

In this study, we assign $H$, $LE$ and $NEE$ as positive when they are directed away from the surface. To avoid negative values of $A_{max}$ and for better readability, we perform sign conversion of $A_{max}$. All statistical analyses and graphing were performed with R version 3.1.1 (R Core Development team, 2014).

**2.5 Multiple Linear Regression Model**

We used a multiple linear regression model ($MLRM$) (Ray-Mukherjee et al., 2014; Whittingham et al., 2006) to investigate the temporal contribution of climatic variables to observed trends in $NEE$. The first $MLRM$ used in this study considers the diel

averaged *NEE*, which includes both the photosynthetic and respiratory processes. We built the model including vapour pressure deficit (*VPD*), atmospheric $CO_2$ concentration (*CO2*), fraction of diffuse radiation (*fdifRad*), wind speed (*wind*), air temperature (*tair*) and actual evapotranspiration divided by potential evapotranspiration (*ET_ET_pot*). Unless otherwise stated, the environmental variables used in this study are measured above the canopy in 22 m height. The form of the model for the 24-hour averaged *NEE* is as follows:

$$NEE = \beta_1 VPD + \beta_2 CO_2 + \beta_3 fdifRad + \beta_4 wind + \beta_5 tair + \beta_6 ET\_ET\_pot \qquad (1)$$

where $\beta$ is the slope. The *MLRM* parameters were estimated using the ordinary least squares method. We transformed each parameter by subtracting the mean and dividing it by the standard deviation. The transformed data has a mean zero with a standard deviation of 1. In the case of the transformed data as well as when an intercept was added in the 24-hour original *NEE* model, temperature and *VPD* became insignificant (p-value >0.5), and thus the goodness of fit decreased by 53 %. Therefore, we did not include the intercept term in equation (1) because without the intercept the model gave a relatively high goodness of fit (see Supplement, Table S1 & Table S2). Initially, we included more parameters for the *MLRM* since we did not put a limit on the number of covariates to explain the observed *NEE*. However, we applied different case scenarios where we examined different *MLRMs* in relation to setting up the model (see sample model case scenarios in the Supplement, Table S2). In these case scenarios we included Akaike information criterion (*AIC*) scores along with the goodness of fit values to ensure the following model criteria: (a) the $\beta$'s are highly statistically significant (Chatfield, 1995), (b) the predictors are chosen in such a way so that they are least correlated (Zuur et al., 2010), and (c) the model has high *AIC* score. In the initial model setup (equation 1) we included drought-indicators such as precipitation and soil moisture at different depths but these predictors were not significant (p-value >0.1). Thus, we excluded them from the model and used only predictors which were highly significant. We also standardized the data to consider normality and non-linearity (Chen et al., 2018), but these changes reduced the goodness of fit by a large amount. Therefore, throughout this study we use the data in the original form.

For the second *MLRM*, we focused on the midday *NEE* (10-14 h local time), which is dominated by photosynthesis and thus avoids any issues of nighttime flux uncertainties. In this case, we used predictors for our model which were significant, i.e. incoming photosynthetically active radiation (*PARin*), *tair*, *VPD*, $CO_2$ and *fdifRad*. The form of the model for the day-time *NEE* is as follows:

$$NEE = \beta_1 PARin + \beta_2 tair + \beta_3 VPD + \beta_4 CO_2 + \beta_5 fdifRad + \beta_6 ET\_ET\_pot \qquad (2)$$

To complement day-time *NEE*, we looked as well at night-time *NEE* (19-5:30 h local time). The modelled *NEE* for the night-time takes the following form:

$$NEE = \beta_1 tair + \beta_2 VPD + \beta_3 ET\_ET\_pot + \beta_4 tair_{12} + \beta_5 wind \qquad (3)$$

For the nighttime *NEE*, we also considered environmental variables within the canopy profile, i.e. air temperature measured at 12 m above the soil (*tair12*). In the night, soil respiration could be influenced by this environmental factor (Zhou et al., 2013). Initially, we also tested the model using soil temperature and soil moisture but these parameters were not significant.

**2.5.1 *NEE* under intensified drought and haze conditions**

We used the above three *NEE* models (equations 1 to 3) based on the 2015-drought and haze conditions to investigate the impacts of intensified non-haze drought (*NHD+*) and haze drought (*HD+*) conditions on oil palm *NEE*. These two scenarios focus on the response of oil palm to short-term more extreme atmospheric conditions associated with projected more severe future *ENSO* events during the current life cycle of the oil palm plantation, which was planted 1999-2004 and is therefore in a

mature stage and in the middle of its life cycle. The temperature change in the scenarios, however, reflects only short-term extreme conditions and does not consider slow long-term effects of a changing climate.

Under intensified non-haze drought (*NHD+*) during the current rotation cycle of the oil palm plantation, we assume a short-term increase in *VPD*, incoming *PAR* and air temperature and a decrease in diffuse radiation. Thus, we modified the mean of the model input variables as *VPD* +20%, *fdifRad* -20%, *tair* +20%, *PARin* +20%, ET_ET_pot -20% and *tair12* +20%. Under

intensified haze drought (*HD+*) we modified the mean of the environmental variables (*VPD* by +20%, $CO_2$ by +20%, *fdifRad* by +20%, *tair* by +20%, *PARin* by -20%, ET_ET_pot -20% and *tair12* by +20%) in the model. For both scenarios (*NHD+* and *HD+*), however, we kept the coefficients of the input parameters constant.

**3 Results**

**3.1 Atmospheric and environmental conditions**

Strong inter-seasonal differences in precipitation pattern, air temperature and atmospheric *VPD* characterize the study period, with the year 2015 being slightly drier and warmer as during the reference periods of 2014 and 2016 (Table 1). From March 2015, both the daily mean air temperature and daily mean *VPD* showed a steady increase and reached their maxima during the haze drought period in mid-October (Figure 2). The first four months in 2015 were cooler and wetter than during the reference period (Table 1). From May until mid-September, when the non-haze drought hit the area in 2015, air temperature and *VPD*

were of similar magnitudes in 2015 and the reference period but accumulated precipitation was as little as 192 mm in 2015 compared to 594 mm during the reference period (Supplement, Figure S1). Inter-seasonal differences in air temperature and in *VPD* were most pronounced from mid-September until mid-November, when haze covered the area in 2015. During that time, mean air temperature was 28.3 ± 0.8°C and mean *VPD* was 8.71 ± 2.57 hPa, which is 2.3°C and 4.98 hPa higher than during the reference period. There were sharp contrasts in soil water content (*SWC*) in 2015 between the pre-drought and haze

drought period due to the absence of precipitation in the latter period. *SWC* in the upper two soil layers (30 & 60 cm) declined by 35%, respectively, while in the bottom layer (100 cm) the decline was 10% (Table 1). During the reference period,

differences in *SWC* were less pronounced, with maximum decline of 26% in the upper two soil layers. Daily mean global radiation and daily mean incoming photosynthetically active radiation (*PAR*) showed strong periodical and day-to-day variations over the course of the study period. In 2015, irradiance reached its maximum during the non-haze drought period in late July and mid-August (Figure 2). After this peak, the continuous emergence of haze led to a substantial decrease in both $R_G$ and *PAR* (Table 1). Simultaneously, fraction of diffuse radiation increased from 0.21 to 0.99 and diffuse radiation remained almost the sole shortwave radiation component for almost two months. Compared to the reference period, daily average incoming *PAR* during the haze drought in 2015 decreased by 107 µmol m$^{-2}$ s$^{-1}$ (-36%) while fraction of diffuse radiation increased by 0.12 (13%) (Table 1). The persistence and density of the haze in 2015 is reflected in daily average sunshine duration (Table 1). During the haze drought period, the sun was, on average, visible for 50 minutes per day, which equals to 7% within 12 hours of potential daylight (sun above the horizon). During the pre-drought, non-haze drought and post-haze period, the sun was visible for 6.7 (56%), 10 (83%) and 6 (50%) hours per day, respectively. Atmospheric $CO_2$ concentration during the haze drought and post-haze period in 2015 was 5% (20 ppm) and 6% (24 ppm) higher than during the reference period.

**3.2 Net ecosystem CO₂-exchange, carbon accumulation and yield**

The oil palm plantation was a net sink of $CO_2$ during the study period. Mainly due to the impact of the haze period, net ecosystem $CO_2$ exchange (*NEE*) in 2015 (-1.79 ± 13.53 µmol m$^{-2}$ s$^{-1}$) was significantly weaker (P <0.01) compared to the reference period (-2.20 ± 14.48 µmol m$^{-2}$ s$^{-1}$) (Table 2). Only in the very beginning of 2015 and during the period June-September 2015, *NEE* was higher compared to the reference period (Figure 3) and $CO_2$ uptake showed a slight increase coinciding with the drought-related increase in incoming *PAR*. The beginning of the haze drought marks a strong transition where $CO_2$ uptake initially decreased with developing haze, followed by a two-month period where the oil palm plantation turned into a small source of $CO_2$ to the atmosphere.

Carbon accumulation by the oil palm plantation was relatively strong in the first months of 2015 and exceeded accumulation of the reference period by up to 80 g C m$^{-2}$ (Figure 3b). During the following months until mid-June, carbon accumulation of the reference period surpassed 2015-carbon accumulation but by mid-August these differences were offset. Due to the haze from October to mid-November 2015, carbon accumulation initially paused, followed by small overall carbon loss of 10 g C m$^{-2}$ within 40 days. After the haze, the oil palm plantation was not able to offset the pause in carbon accumulation and carbon losses during the haze and therefore, the total amount of accumulated carbon in 2015 was 152.7 g C m$^{-2}$ (18%) lower compared to the reference period (Table 1).

Over the course of the non-haze drought, the oil palm plantation reduced its maximum rate of photosynthesis ($A_{max}$) (Figure 4). However, drought-related changes in meteorological and environmental conditions caused a minor (3%) decrease in $A_{max}$ compared to pre-drought conditions. With the continuous development of haze in September 2015 and related absence of direct sunlight the oil palm plantation seemed to compensate for the overall haze-related reduction in incoming *PAR*, with a jump of

$A_{max}$ by 13 µmol m$^{-2}$ s$^{-1}$ (37%) within a couple of days (Figure 4). This compensation effect of relatively high $A_{max}$ continued over the haze drought period, with $A_{max}$ being 4.8 µmol m$^{-2}$ s$^{-1}$ (18%) higher than during the non-haze drought.

Using linear regression between monthly $NEE$ and oil palm yield, we found that a 6-month delay in yield showed highest R$^2$ of 0.36 (P<0.01) with $NEE$. Therefore, we consider the period November 2015 to May 2016 as the time when $NEE$ and carbon accumulation during the non-haze drought and haze drought in 2015 were reflected in monthly oil palm yield. From August 2015, monthly oil palm yield declined continuously from 3.93 t ha$^{-1}$ to its minimum of 1.05 t ha$^{-1}$ in May 2016. Compared to the same period (Nov.-May) in the two years before and the year after the $ENSO$ event, average yield affected by 2015-drought and haze was 32% (0.70 t ha$^{-1}$) lower. Considering the 2015-haze drought only, average oil palm yield 6-9 months after the beginning of the haze drought was even 50% (1.1 t ha$^{-1}$) lower compared to the non-$ENSO$ years.

## 3.3 Evapotranspiration and turbulent heat fluxes

Total evapotranspiration ($ET$) derived from $EC$ latent heat flux ($LE$) measurements was 1245 ± 362 mm yr$^{-1}$ in 2015 and 1580 ± 469 mm yr$^{-1}$ during the reference period (Table 2), with a higher share of $ET$ on precipitation during the reference period (77.9%) compared to 2015 (64.5%). During the non-haze drought and haze drought periods, the oil palm plantation experienced strong water loss from $ET$ as $ET$ was 2.5 and 1.2 times the amount of precipitation, respectively. $ET$ was lowest during the haze drought period (Figure 5, Table 2), mainly driven by the reduction in incoming solar radiation and $PAR$ as well as by oil palm drought and heat stress which may have triggered partial stomata closure, especially in the beginning of the haze drought when $VPD$ was high (Figure 2). Conversely, partial stomata closure during high $VPD$ as well as the absence of precipitation and related drying of the upper soil generally increased sensible heat fluxes ($H$) at the cost of $LE$ and $ET$, reflected in the behaviour of the Bowen ratio ($H/LE$) (Figure 5). From the first half of the pre-drought period into the second half of the non-haze period, the Bowen ratio showed a steady but relatively small decline. However, the end of the non-haze drought and the beginning of the haze drought period mark a strong transition in the behaviour of the Bowen ratio, manifested by a strong jump, peak values of ~0.38 and average of 0.25 for approx. one month. This jump in the Bowen ratio might be related to the increasing density of the haze and related reduction in incoming $PAR$ in combination with high $VPD$ which decrease $LE$ mainly via oil palm water and light stress to a greater extent than the general drying of the soil and lack of precipitation.

## 3.4 Drivers of net ecosystem CO$_2$-exchange

Modelled $NEE$ from our $MLRM$ simulated a small positive effect on $NEE$ during the non-haze drought, with an increase in CO$_2$ uptake by 0.32 µmol m$^{-2}$ s$^{-1}$, and a negative effect on $NEE$ during the haze drought, with a decrease in CO$_2$ uptake by 0.99 µmol m$^{-2}$ s$^{-1}$ (Figure 6, Supplement Table S5). Modelled $NEE$ is in good agreement with the measured $NEE$, i.e. for midday (10-14 h local time), nighttime (19-5:30 h) and average $NEE$ (0-24 h) the model explains 98%, 94% and 83%, respectively, of the temporal variability in the measured $NEE$. Overall, the relative change of meteorological and environmental parameters during the non-haze drought and haze drought caused a more pronounced response of $NEE$ in the latter period compared to non-drought and non-haze conditions, especially during midday (Figure 6).

During the non-haze drought, changes in radiation components were the main predictors of changes in midday $NEE$. Higher incoming $PAR$ increased $CO_2$ uptake while at the same time, this gain in $CO_2$ uptake was compensated by the negative impact of decreasing fraction of diffuse radiation (Figure 6, Supplement Table S5). However, this estimated effect of the changes in irradiance on $NEE$ was clearly small compared to the negative effects of dim light conditions during the haze drought where a reduction in incoming $PAR$ resulted in strong decrease in $CO_2$ uptake (Figure 6). Further, the effect of incoming $PAR$ and fraction of diffuse radiation on midday $NEE$ was reversed during the haze drought compared to the non-haze drought and the decrease in fraction of diffuse radiation contributed to higher midday $CO_2$ uptake but these positive effects were almost offset completely by the decrease in incoming $PAR$.

Increasing $VPD$ had a negative impact on midday $NEE$ (decrease in $CO_2$ uptake), while the increase in air temperature had a positive impact on midday $NEE$ (increase in $CO_2$ uptake). Oil palm drought stress, manifested in a general decrease in $ET/ET_{pot}$ (Table 2), was less severe during the non-haze drought compared to the haze drought period, resulting in a slightly more pronounced decrease in $CO_2$ uptake during the latter period (Figure 6). The observed changes in atmospheric $CO_2$ concentrations during the non-haze drought and haze drought suggest that the oil palm might respond via photosynthesis and stomata behaviour to the elevated atmospheric $CO_2$ levels. However, rising atmospheric $CO_2$ concentration had no fertilization effect for the oil palm plantation, in contrary, the increase in $CO_2$ concentration contributed to a decrease in $CO_2$ uptake (Figure 6).

During both non-haze drought and haze drought, the change in nighttime (19-5:30 h) air temperature above the canopy was the main predictor of changes in nighttime $NEE$ (respiration). The increase in air temperature tended to increase respiration. This was more pronounced during the haze drought compared to the non-haze drought (Figure 6, Supplement Table S5 & S6).

**3.5 $NEE$ under intensified drought and haze conditions**

Our two model projections, where we increased the effects of non-haze drought and haze drought conditions based on the 2015-drought and haze conditions, showed that increased non-haze drought conditions ($NHD+$) enhanced $CO_2$ uptake while increased haze drought ($HD+$) weakened $CO_2$ uptake and might even promote $CO_2$ release (Figure 7, Supplement Table S7). Daily average (24-hour) $CO_2$ uptake in $NHD+$ was increased by 2.25 µmol m$^{-2}$ s$^{-1}$ compared to the 2015-non-haze drought conditions. $NHD+$ might enhance midday $CO_2$ uptake and nighttime respiration, which increased by 6.52 µmol m$^{-2}$ s$^{-1}$, 1.59 µmol m$^{-2}$ s$^{-1}$, respectively, mainly due to the effect of a high air temperature in $NHD+$ which is also the main predictor of daily average, midday and nighttime $NEE$ (Supplement Table S7). Incoming $PAR$ is the dominant light parameter for $NEE$ and increases in incoming $PAR$ in $NHD+$ are able to offset the modelled negative impact of decreased fraction of diffuse radiation on $NEE$. This is contrary to what the model suggested for the 2015-non-haze drought reference conditions where we observe that the increase in incoming $PAR$ was not able to offset the negative impacts on $NEE$ due to decreased fraction of diffuse radiation. Similar to $NHD+$, air temperature in the increased haze drought scenario ($HD+$) was the main predictor of $NEE$ and contributed to a high midday and daily average (24-hour) $CO_2$ uptake and also to a high nighttime respiration (Figure 7, Supplement, Table S8). However, the negative effects of $HD+$ offset the positive effects of increased air temperature. Daily

average (24-hour) $CO_2$ uptake and midday $CO_2$ uptake in *HD+* were decreased by 0.85 µmol m$^{-2}$ s$^{-1}$, 4.51 µmol m$^{-2}$ s$^{-1}$, respectively, while nighttime ecosystem respiration was increased by 2.53 µmol m$^{-2}$ s$^{-1}$. Incoming *PAR* in *HD+* remains the dominant light parameter on midday *NEE* and its decrease cannot be offset by the positive effects of increased fraction of diffuse radiation. In *HD+*, midday *VPD* is of less relative importance on *NEE* as compared to the reference haze drought

conditions. As already observed in the 2015-haze drought model output, increased $CO_2$ concentration in *HD+* does not act as an additional fertilization for the oil palm plantation. In contrast, the negative impact of increased $CO_2$ concentration on *NEE* becomes the dominant predictor of *NEE* in *HD+*. Our two scenarios indicate that increased drought stress, reflected by decreasing *ET/ET$_{pot}$*, has more pronounced negative impact on *NEE* in *HD+* compared to *NHD+*. However, oil palm seems to be relatively resistant against drought since the overall impact of changes in *ET/ET$_{pot}$* on *NEE* was relatively small in both

scenarios.

## 4 Discussion

### 4.1 Oil palm response to drought and haze conditions

Oil palm has exceptionally high photosynthetic efficiency compared to most of the vascular plants (Apichatmeta et al., 2017) but this efficiency comes with a downside: Oil palm, like many other tropical plants, shows a distinct reaction to changes in

atmospheric and soil parameters, with gradual symptoms of water and heat stress which directly affect photosynthesis and evapotranspiration as well as fruit bunch development and yield (Bakoumé et al., 2013; Paterson et al., 2013). During our study period, we observed that accumulated annual precipitation 2015 and during the reference period was on the lower limit of reported optimum precipitation range for oil palm (Pirker et al., 2016). However, oil palm requires minimum precipitation of 100 mm per month to avoid drought stress (Corley & Tinker, 2016). This was not fulfilled in September 2014, from June

to October 2015, and in January 2016. Previous studies report a strong correlation between *NEE* and soil moisture (Méndez et al., 2012; Cha-um et al., 2013), with declining $CO_2$ assimilation under dry conditions. In our study, however, we found no strong correlation between *NEE* and soil moisture conditions, and between *NEE* and *ET/ET$_{pot}$* during the non-haze drought and haze drought period. This might be explained by the relatively stable soil moisture conditions in deeper layers (100 cm) of the oil palm plantation which, compared to the upper layers (30 and 60 cm) showed only a moderate decrease during both non-

haze drought and haze drought (Table 1). Oil palm seems to be able to uptake water from deep soil and store the water in the trunk during night, supporting water use during peak hours of photosynthesis (Niu et al., 2015; Meijide et al., 2017). Therefore, the relatively moderate decrease in soil moisture in deeper soil layers might have had a limited effect on *NEE*. Temperature increase and related heat stress is another factor which might negatively affect the growth of oil palm (Oettli et al., 2018). Our analysis did not support this finding because during the non-haze drought the effect of increasing temperature

on *NEE* was almost negligible. During the haze drought, higher air temperature had a positive impact on $CO_2$ uptake although the haze period experienced the highest air temperature during the entire study period. Changes in temperature and moisture availability also impact oil palm ecosystem respiration. Matysek et al. (2018) observed high heterotrophic carbon loss from

drained peat soils in a Malaysian oil palm plantation during the dry season and Sigau & Hamid (2018) found similar behaviour in Malaysian rubber and oil palm plantations on drying Haplic Nitisols soils but both studies report only minor impact of increased soil temperature on soil respiration. Autotrophic respiration, however, tends to decrease with increasing leaf temperature (Slot et al., 2014). In our study, the increase in air temperature tended to increase nighttime ecosystem respiration

and therefore might also lead to higher day time respiration during the non-haze drought and haze drought period.

Oil palm, such as other tropical plant species, seems particularly susceptible to changes in atmospheric *VPD* (Dufrene & Saugier, 1993; Cunningham, 2005; Lamade & Bouillet, 2005; Wahid et al., 2005; Bayona-Rodríguez & Romero, 2016; Mathews & Ardiyanto, 2016) with high levels of *VPD* causing partial closure of stomata and limiting photosynthesis and transpiration. Our *MLRM* and measurements are in line with these findings and high levels of *VPD* had a stronger impact on

*NEE* during the haze drought period compared to the non-haze drought period. To a certain extent, oil palm is capable to adjust its stomatal regulation to short-term periods of moderate *VPD* and soil water deficit by increasing its maximum rate of photosynthesis ($A_{max}$) (Dufrene & Saugier, 1993; Apichatmeta et al., 2017). However, during the non-haze drought and haze drought those two environmental parameters exerted only little impact on $A_{max}$ and changes in irradiance seemed to be the dominant driver of $A_{max}$.

Oil palm grows in regions with high solar flux densities (Barcelos et al., 2015) and it is able to strategically optimize its photosynthesis to light conditions, with pronounced diurnal effects and maximum efficiency before or at about midday (Apichatmeta et al., 2017). In our study, measurements and *MLRM*-results showed strongest response of oil palm *NEE* to drought, haze and changes in irradiance during midday. Due to the reduction of incoming *PAR* for almost two months, the haze was a major and persistent disturbing factor for oil palm *NEE* and $A_{max}$. The initial increase in diffuse light conditions and

its positive impact on $A_{max}$ and *NEE* cannot compensate for the reduction in incoming *PAR*. Therefore, the observed pause in carbon accumulation and even small carbon release during the haze drought could have been prevented since without the haze, the oil palm plantation would have remained a sink of $CO_2$ during that period.

Changes in oil palm yield are one direct consequence of varying nutrient, meteorological and climatic conditions (Sun et al., 2011; Mathews & Ardiyanto, 2016; Oettli et al., 2018). Prolonged drought and nutrient limitation does not only affect carbon

accumulation via photosynthesis but leads to abortion of female inflorescences and failing bunch yield (Bakoumé et al., 2013). In an oil palm plantation in Central Kalimantan (Indonesia) dense haze from peat fires resulted in poor quality of the fruit bunches and in low oil palm extraction rates (Mathews & Ardiyanto, 2016). Fertilization under water stress conditions has negative impact on oil palm growth and may further reduce oil palm yield while fertilization during well-watered conditions promotes oil palm growth and yield (Sun et al., 2011). At our study site, fertilizers are applied at the end of the wet season

(April-May) and in 2015, precipitation was still sufficiently high to maintain well-watered soil conditions during the fertilization. Oil palm yield in 2016, and its initial sharp drop by the end of 2015 can therefore be related to the drought and haze conditions and the haze was the driving stressor. Similar to the effects of haze on *NEE*, without the haze oil palm yield might not have experienced such a sharp decline.

Short-term elevated $CO_2$ exposure on oil palm seedlings (Ibrahim et al., 2010; Jaafar & Ibrahim, 2012) and on mature oil palm (Henson & Harun, 2005) have shown that elevated $CO_2$ concentration promote plant growth due to elevated rates of photosynthesis and reduced water loss by transpiration. To our knowledge, no comprehensive study has investigated the complex interplay of elevated $CO_2$ concentrations, increased temperature and decrease in radiation in oil palm. Mathews &
Ardiyanto (2016) speculate that short-term elevated levels of $CO_2$ under haze conditions and related potentially strong stomatal opening may offset for the lack of irradiance and related shorter timing of stomatal opening. Based on leaf gas exchange measurements in trees, Urban et al., (2014) come to a contradiction that low irradiance is incapable to activate stomatal opening since plants exposed to elevated $CO_2$ levels require higher stomatal activation energy. From our results, it is highly doubtable that elevated $CO_2$ exposure during the haze had any fertilization effect. On the contrary, increasing atmospheric $CO_2$
concentration acted as an additional stress factor for oil palm and decreased $CO_2$ uptake.

Ground-level ozone exerts strong toxicity on tropical and sub-tropical agricultural and natural vegetation (Moraes et al., 2004; Felzer et al., 2007; Zhang et al., 2014; Chen et al., 2018). Ozone concentration was not measured in this study but biomass burning (Kita et al., 2000), as well as nitrogen management and isoprene emissions in oil palm plantations (Hewitt et al., 2009 & 2011) are considered to significantly affect near-surface ozone concentration due to emission of ozone precursor gases. Fire
air pollution generally leads to a decrease in gross primary productivity (*GPP*) (Yue & Unger, 2010). To our knowledge, no study has focused on ozone concentration from biomass burning during the 2015 *ENSO* event but studies observe a strong increase in ozone concentration from biomass burning during the 1997-*ENSO* (Thompson et al., 2001) and during the 2006-*ENSO* event (Nassar et al., 2009). At our study site, we therefore expect an increase in ground-level ozone concentration during the haze drought period which might have negatively affected oil palm carbon sequestration.

Increased aerosol concentration from biomass burning and related increase in diffuse light increase plant photosynthesis and therefore decrease the ratio of sensible to latent heat (Steiner et al., 2013). However, in our study and during the peak of the drought when forest fires started to develop in the area, we observed increase in the ratio of sensible to latent heat (Bowen ratio) which is likely due to water stress and related partial stomata closure at high *VPD* (Dufrene & Saugier, 1993; Oettli et al., 2018).

Further, increased aerosol concentration is able to increase overall canopy photosynthesis under moderately enhanced diffuse light conditions (Knohl et al., 2008; Mercado et al., 2009; Kanniah et al., 2012) and sun-exposed leaves seem to benefit from lower *VPD* while shaded leaves benefit from increased diffuse light conditions (Wang et al., 2018). Although our measurements and *MLRM* suggest that the leaves benefitted from the increase in diffuse light conditions during the haze drought period, the high level of *VPD*, especially during midday, was an overall stress factor for the oil palm plantation and
therefore resulted in a decrease in $CO_2$ uptake. At our study site, increased fraction of diffuse radiation due to biomass burning had an overall positive impact (increase in $CO_2$ uptake) and decreased incoming *PAR* a negative impact on $CO_2$ uptake, which is in line with the findings of Malavelle et al (2019). However, while the authors of that study conclude that the positive impact of increased diffuse light conditions offsets the negative impact of decreased *PAR* we observe that the increase in diffuse light conditions is not able to offset the negative impact in decreased *PAR*. We suggest that the strong intensity and relatively long

duration of the haze, with persistently high values of fraction of diffuse radiation for approx. two months, exceeded an optimal range of diffuse fraction (Knohl et al., 2008) and therefore inhibited a positive impact on $CO_2$ uptake.

## 4.2 Short-term response of oil palm to changed climatic conditions and adaptation strategies

Paterson et al., (2015) report that increasing frequency of drought in Southeast Asia has already caused a decline of 10-30% in palm oil production. Our study supports the findings of Dufrene & Saugier (1993) and Apichatmeta et al. (2017) that short-term drought conditions and elevated irradiance under the current or potentially amplified ENSO conditions may be beneficial for oil palm growth since we observe an increase in $CO_2$ uptake during the non-haze drought despite relatively high *VPD* and low soil moisture content. Our scenario of increased non-haze drought (*NHD+*) suggests that drought conditions may enhance $CO_2$ uptake to a certain extent, mainly due to increased incoming *PAR* and increased air temperature. However, our scenario does not consider a temporal prolongation of the drought or a constant increase in temperature associated with elevated temperatures as a result of global rising $CO_2$ levels. We only considered changes in the magnitude of the atmospheric and environmental parameters under the current climate conditions which we expect to be rather constant for the current life cycle of the oil palm plantation. Therefore, we cannot rule out that this modelled positive effect of *NHD+* on $CO_2$ uptake can be maintained if drought conditions remain over a longer period but the relatively weak impact of $ET/ET_{pot}$ on *NEE* suggests that oil palm is relatively resistant to drought.

The reduced irradiance due to fire-induced haze is another stressor for oil palm since it occurs during those periods when the oil palm plantation is already negatively affected by drought and heat. Similar to *NHD+*, we did not include temporal changes in the length of the increased haze drought scenario (*HD+*) but we see that *HD+* may amplify the negative impacts on oil palm *NEE*. Changes in ozone and aerosol concentrations caused by biomass burning have not been measured in our study but it is very likely that both had an additional negative impact on *NEE* (decrease in $CO_2$ uptake) which we are quantitatively not able to capture with our *MLRM*. Nevertheless, negative impacts of *ENSO*-related droughts on oil palm productivity, carbon sequestration, growth and yield are strongly coupled with the temporal and spatial occurrence of fire-induced haze and its ancillary effects such as reduced incoming *PAR*, as well as air pollution of increased ozone and aerosol concentration.

It has been shown that fertilized mature commercial oil palm plantations transpire more water than tropical rainforests due to high productivity (Manoli et al., 2018), thus making them more prone to the effects of droughts (Bakoumé et al., 2013). Adaptation strategies, such as short-term irrigation or the establishment of irrigation ditches may dampen the drought-related impacts in oil palm plantations but aggravate the depletion of natural water reservoirs (Manoli et al., 2018). Elongated periods of drought, as shown in this study, increase sensible heating at the cost of evapotranspiration, resulting in surface warming. Oil palm plantations have a strong potential to further amplify air heating during droughts since they are hotter and dryer as compared to tropical rainforest and rubber monocultures even during non-El Niño years (Hardwick et al., 2015; Meijide et al., 2018). Covering vast areas of tropical lowlands of Sumatra and Borneo, oil palm plantations have already caused an increase in land surface temperature (Sabajo et al., 2017).

State-of-the-art process-based land surface schemes, such as the Community Land Model (*CLM4*.5) (Oleson et al., 2013; Fan et al., 2015), are powerful tools to address ecosystem surface energy balance, hydrological processes and carbon-nitrogen biogeochemistry (Oleson et al., 2013; Fan et al., 2015). Although these models are well-developed and widely-used, they fail to include smoke haze as an environmental parameter affecting ecosystem behaviour. In this study, we used a simple multi linear regression model (*MLRM*) to assess the impact of haze drought on oil palm productivity and developed an increased haze scenario (*HD*+). With this simple model we were able to show strong site-specific negative response of oil palm to haze drought. These specific results of oil palm behaviour during drought and haze conditions might be useful to parameterize models, such as *CLM* and even applicable to other ecosystem and land-use types.

## 5 Conclusions

In this study, we investigate the impact of drought and smoke haze on the net $CO_2$ exchange, evapotranspiration, yield and surface energy budget in a commercial oil palm plantation. We found that drought and smoke haze conditions, with related increase in atmospheric *VPD* and air temperature, and changes in light conditions are major disturbing factors for the oil palm plantation. Our measurements and *MLRM* showed that the strong haze amplified the negative effects of the drought. It is very likely that without the haze, the negative impact on $CO_2$ fluxes, carbon accumulation and yield would have been less pronounced. Although micrometeorological measurements in oil palm plantations become more and more frequent, there is still a substantial lack of air quality measurements, e.g. ozone or aerosol concentration. In our study, smoke haze may have substantially increased ozone and aerosol concentration which both further negatively impact the oil palm plantation. Fire-preventing measures such as sustainable land management, stricter law enforcement and sanctioning, strategic hazard planning and awareness-raising on the effects of fires on oil palm yield but also on air quality and health may help to mitigate the negative effects of drought. Further, incorporating smoke haze as an environmental stress factor into regional or global model approaches may foster more accurate estimations of ecosystem $CO_2$, energy and water vapour flux behaviour during such extreme meteorological events and may allow a more holistic viewpoint of possible adaptation strategies and hazard-prevention caused by *ENSO*.

## Code and data availability

The code and data used in this study are available on GitHub (https://github.com/CbioST/ENSO_OilPalm).

## Author contribution

The original idea of the paper was suggested by Alexander Knohl (AK), Ana Meijide (AM) and Christian Stiegler (CS) and discussed and developed by all authors. AM performed the field work and CS performed the data analysis. Ashehad Ashween

Ali (AA) and CS developed the model code, run the simulations and performed the model analysis. CS prepared the manuscript with contributions from all co-authors.

**Acknowledgements**

This study was funded by the Deutsche Forschungsgemeinschaft (DFG, German Research Foundation) – project number 192626868 – SFB 990 and the Ministry of Research, Technology and Higher Education (Ristekdikti) in the framework of the collaborative German - Indonesian research project CRC990, subproject A03 and A07. The authors wish to thank our local field assistants in Indonesia, i.e. Basri, Bayu and Darwis as well as Edgar Tunsch, Malte Puhan, Frank Tiedemann and Dietmar Fellert for technical support. We also thank PTPN6 for permission to conduct our research at the oil palm plantation.

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

**Figures**

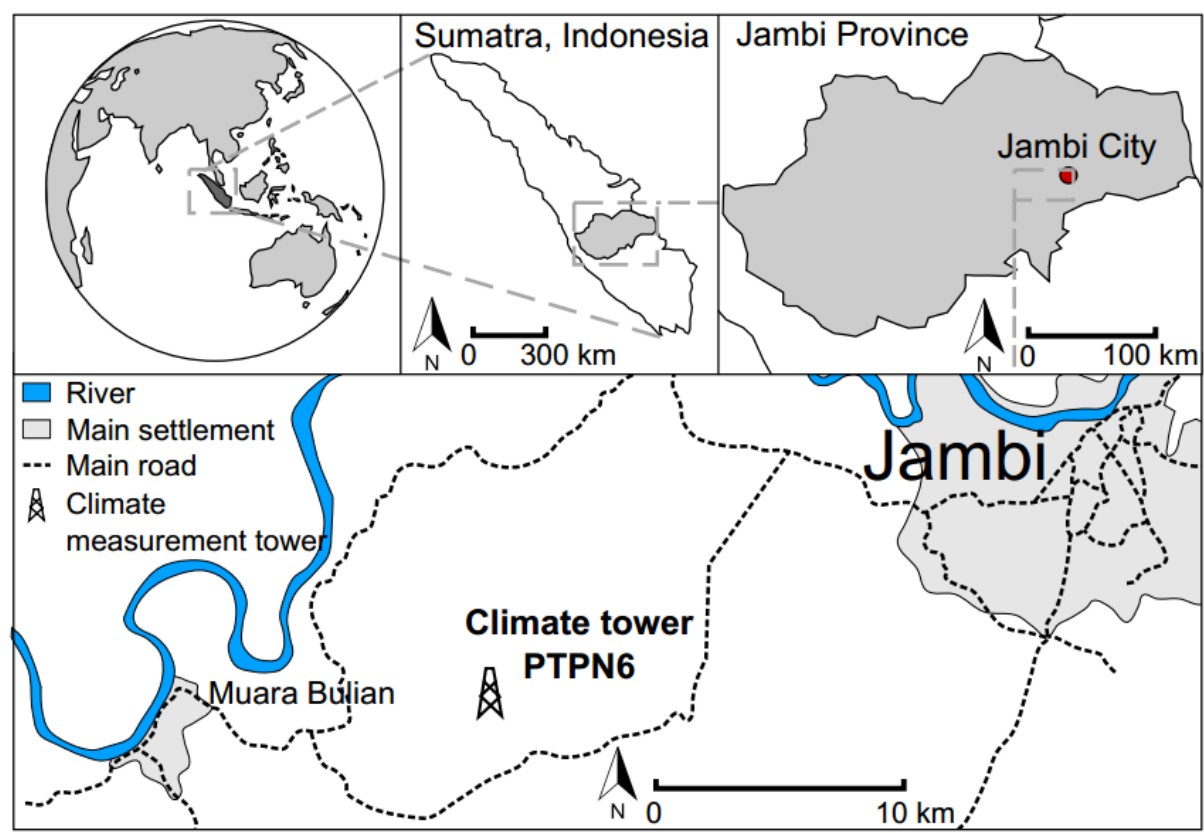

**Figure 1: Map and location of the study site and climate measurement tower at PTPN6 oil palm plantation, approx. 15 km south-west of the city of Jambi (Sumatra, Indonesia)**

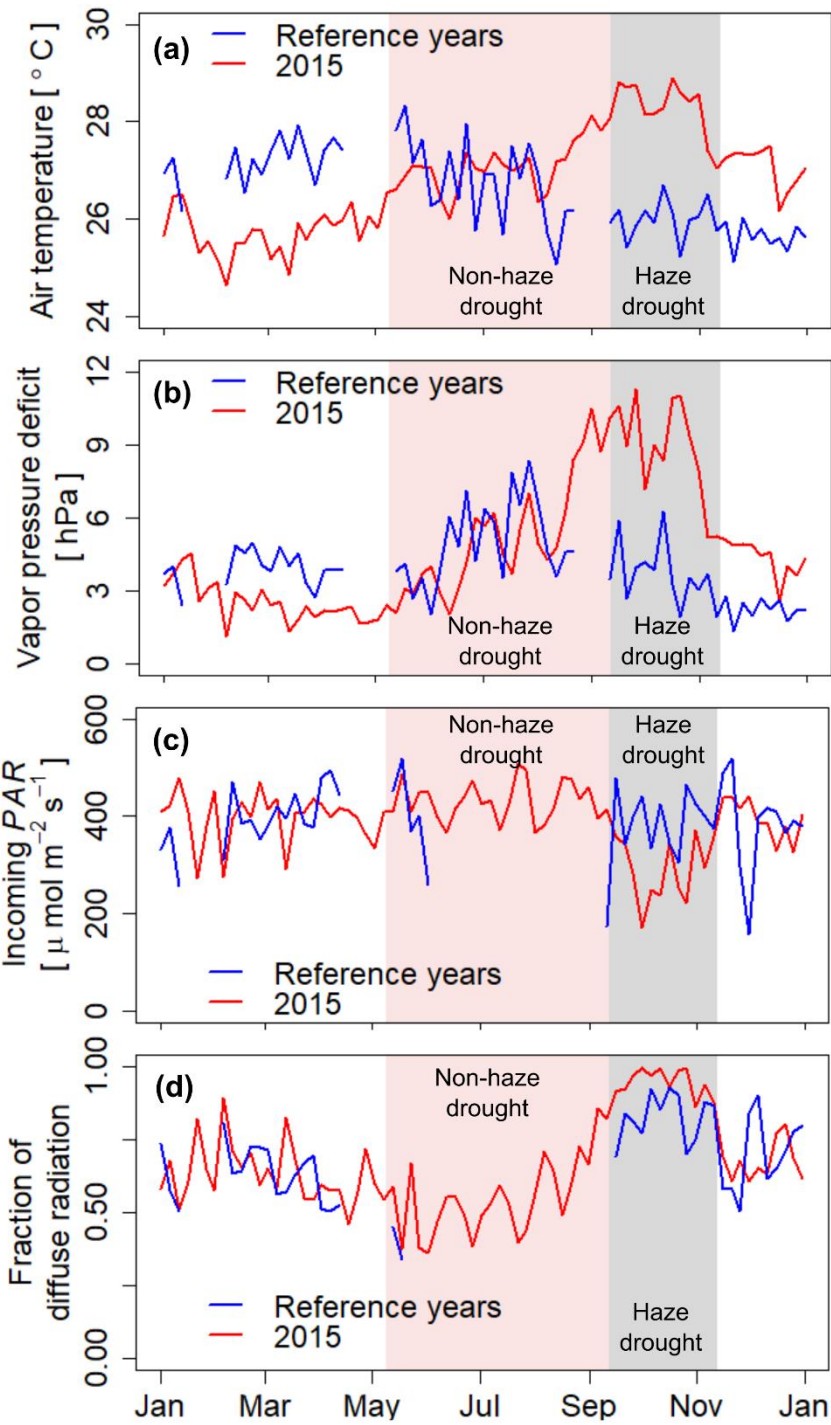

**Figure 2: Five-day running mean of air temperature (a), atmospheric vapour pressure deficit (*VPD*) (b), incoming photosynthetically active radiation (*PAR*) (c), and fraction of diffuse radiation (d) during 2015 and the reference time period. Shaded areas in red and grey mark the non-haze drought and the haze drought period in 2015, respectively.**

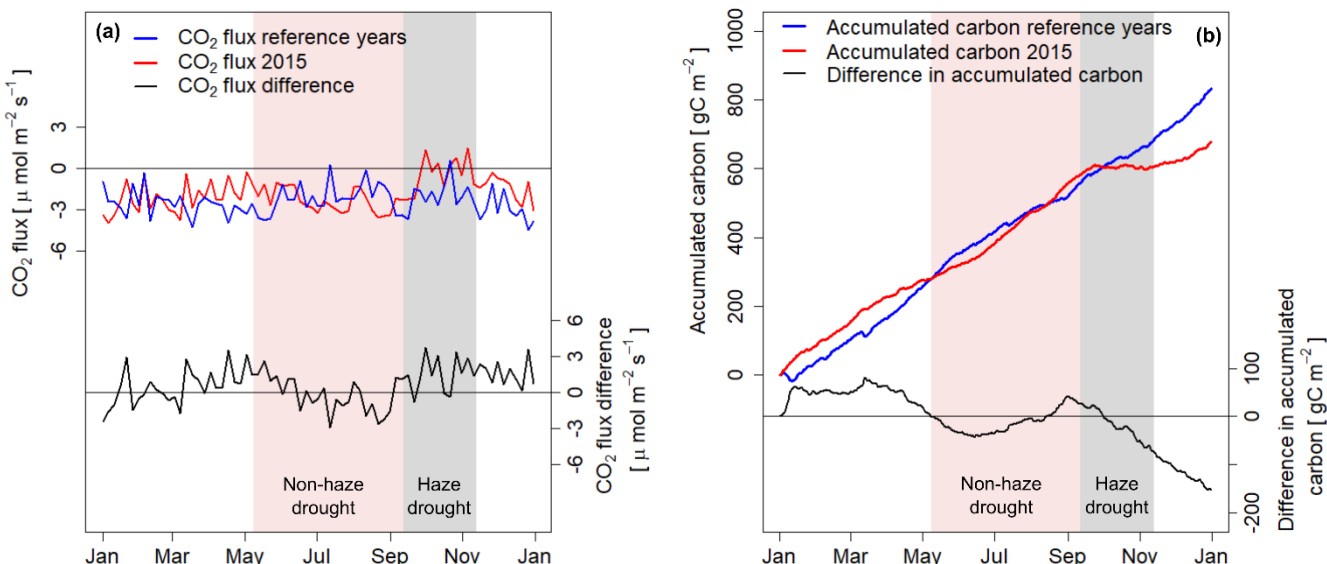

**Figure 3: (a)** Five-day running mean of net ecosystem $CO_2$-exchange (*NEE*) during 2015 and the reference time period and five-day running mean of $CO_2$ flux difference (2015 minus reference time period). **(b)** Accumulated carbon uptake derived from $CO_2$ fluxes during the period 2015 and the reference time period, and differences in accumulated carbon between the two periods (2015 minus reference time period). Shaded areas in red and grey mark the non-haze drought and the haze drought period in 2015, respectively.

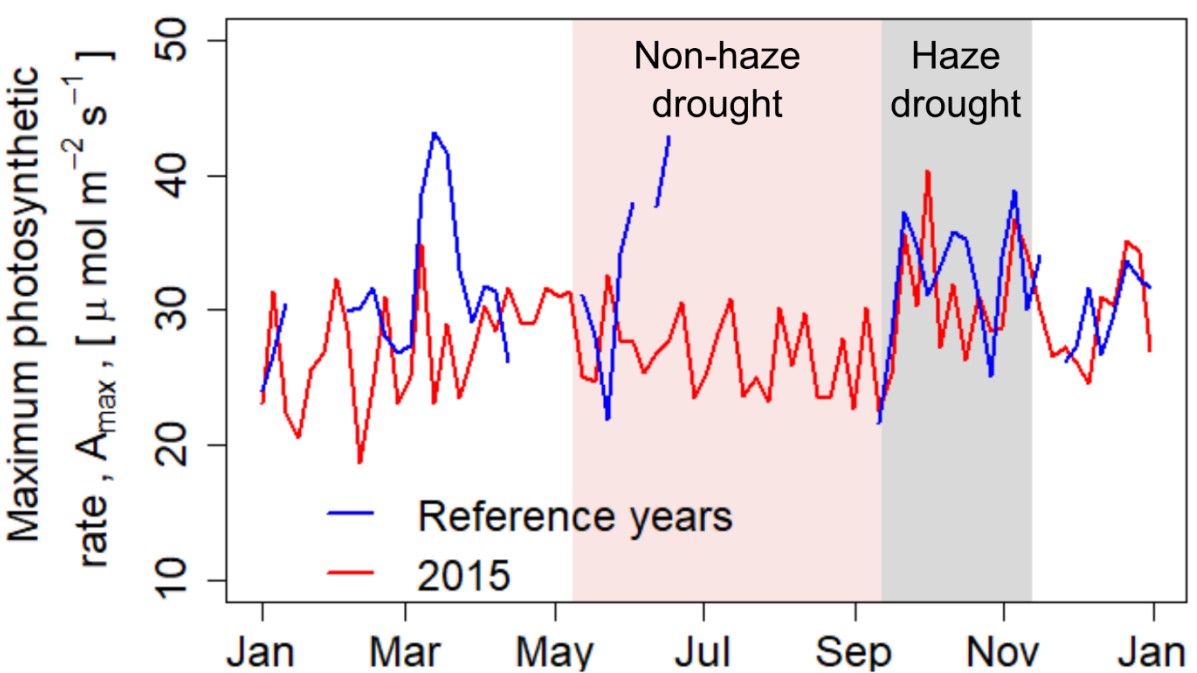

**Figure 4:** Five-day running mean of maximum rate of photosynthesis ($A_{max}$) during 2015 and during the reference time period. Sign convention has been performed to avoid negative values of $A_{max}$. Shaded areas in red and grey mark the non-haze drought and the haze drought period in 2015, respectively.

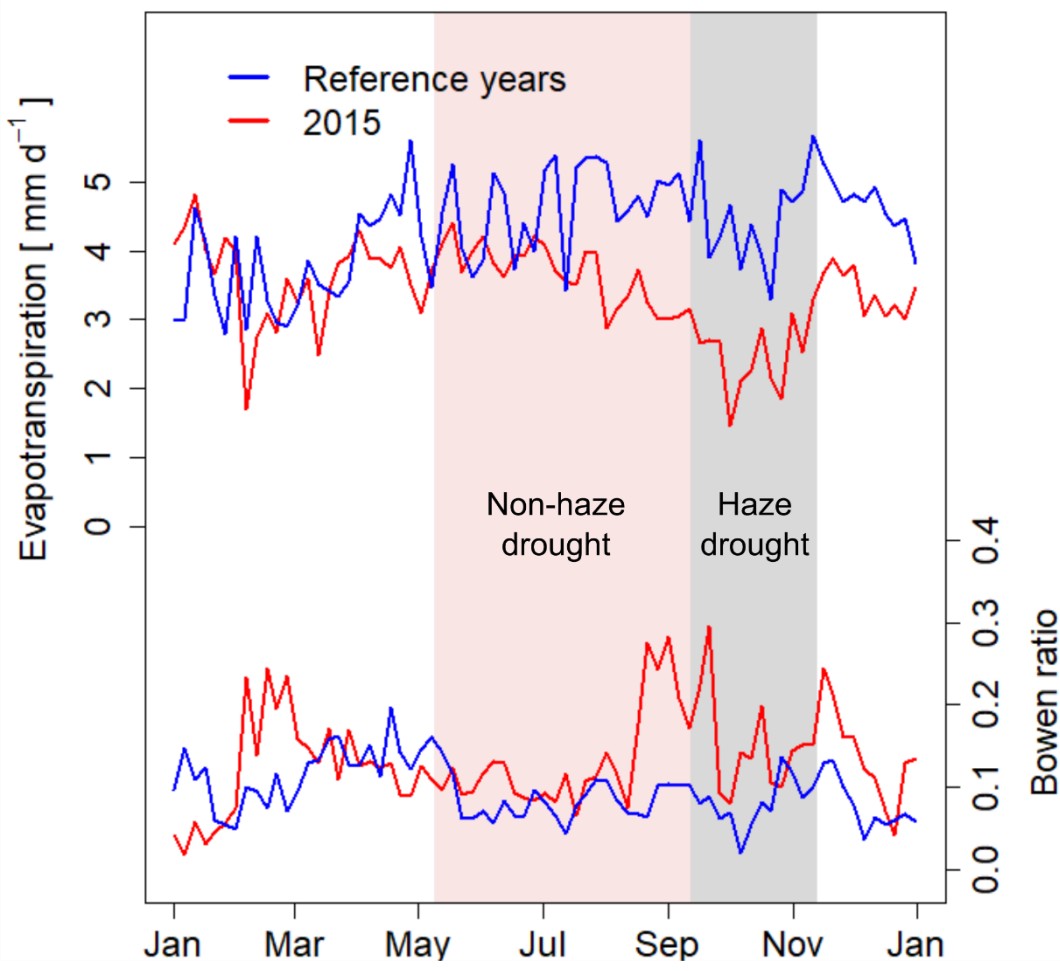

**Figure 5: Five-day running mean of daily evapotranspiration and ratio of sensible to latent heat fluxes (Bowen ratio) during 2015. Shaded areas in red and grey mark the non-haze drought and the haze drought period in 2015, respectively.**

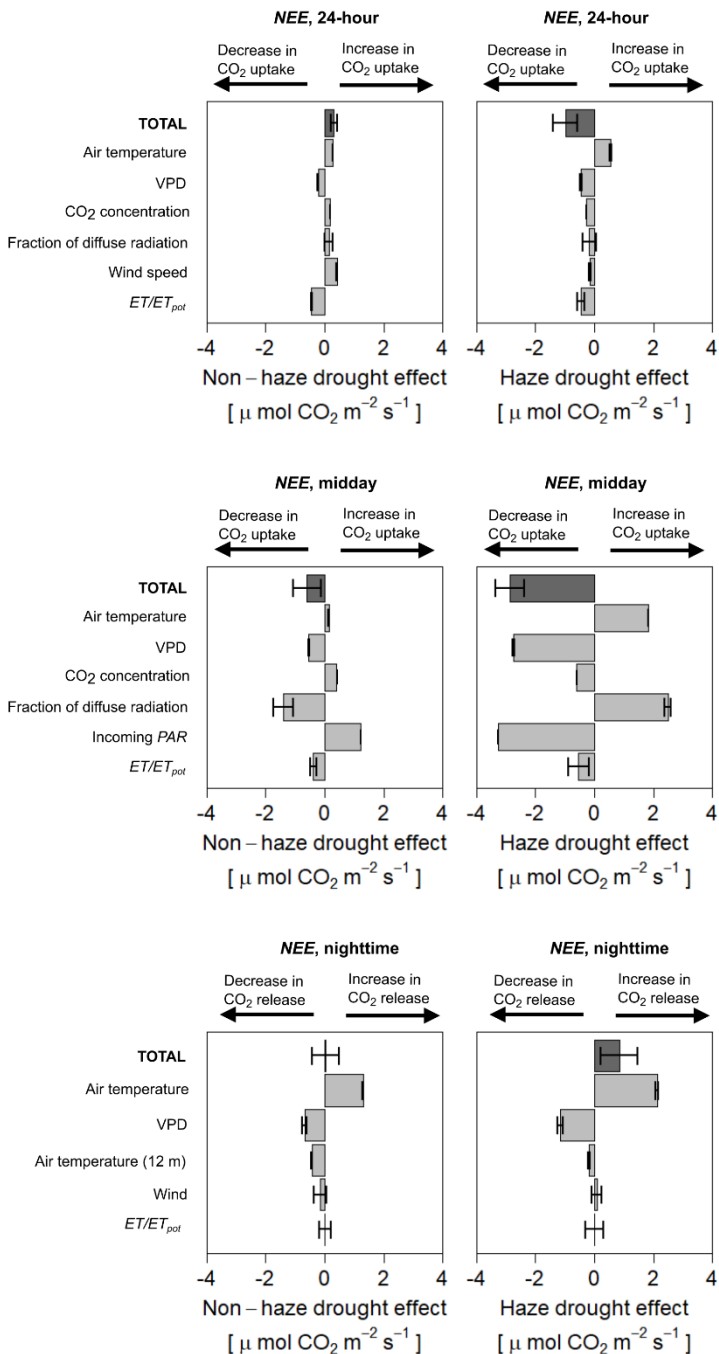

**Figure 6: Contribution and effect of meteorological and environmental parameters during the non-haze drought and haze drought period on 24-hour (upper), midday (middle) and nighttime (lower) net ecosystem $CO_2$ exchange (*NEE*) compared to non-drought and non-haze conditions using Multiple Linear Regression Model (*MLRM*). Error bars show the standard error.**

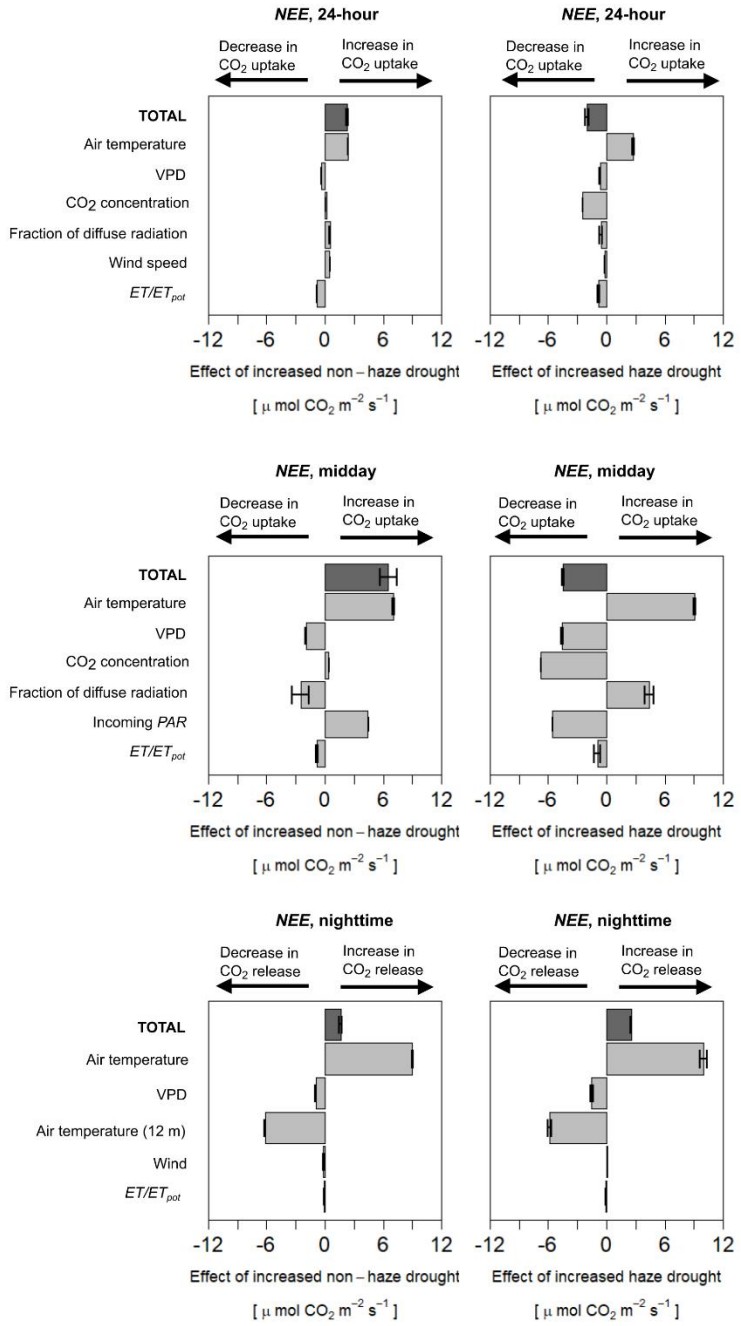

**Figure 7: Contribution and effect of meteorological and environmental parameters considering increased non-haze drought (*NHD+*) and increased haze drought (*HD+*) scenario on 24-hour (upper), midday (middle) and nighttime (lower) net ecosystem CO₂ exchange (*NEE*) using Multiple Linear Regression Model (*MLRM*). Error bars show the standard error.**

**Tables**

Table 1: Meteorological parameters (daily mean ± SD, or accumulated for precipitation and carbon) derived from 30-minute averages or sums during pre-drought, non-haze drought, haze drought and post-haze period in 2015, for the entire year 2015 and the reference period (May 2014-December 2014, January 2016-July 2016).

| Period | Air temperature [°C] | Precipitation [mm] | Vapour pressure deficit (VPD) [hPa] | Soil moisture, 30 cm depth [vol%] | Soil moisture, 60 cm depth [vol%] | Soil moisture, 100 cm depth [vol%] | Incoming PAR [µmol m$^2$s$^{-1}$] | Fraction of diffuse radiation | Sunshine duration [hours d$^{-1}$] |
|---|---|---|---|---|---|---|---|---|---|
| Pre-drought (128 days) | 25.7 ± 0.7 | 1003 | 2.53 ± 1.25 | 32.5 ± 1.8 | 31.9 ± 1.4 | 32.3 ±0.8 | 396.9 ± 105.0 | 0.67 ± 0.19 | 6.7 ± 6.9 |
| Drought (127 days) | 27.1 ± 0.7 | 192 | 5.30 ± 2.60 | 27.9 ± 4.3 [A)] | 26.8 ±4.3 [B)] | 27.1 ±2.9 | 432.0 ± 70.6 | 0.57 ± 0.18 | 10.0 ± 7.1 |

| | | | | | | | | |
|---|---|---|---|---|---|---|---|---|
| Haze (61 days) | 28.3 ± 0.8 | 127 | 8.71 ± 2.57 | 18.1 ± 1.5 | 17.5 ± 0.2 | 24.4 ±0.1 | 293.2 ±97.3 | 0.95 ± 0.07 | 0.8 ± 3.2 |
| Post-haze (49 days) | 27.1 ± 0.9 | 608 | 4.30 ± 1.45 | 23.4 ± 1.3 [C] | 20.6 ±1.9 [C] | 26.8 ±2.1 | 393.8 ± 111.0 | 0.71 ± 0.17 | 6.0 ± 6.8 |
| 2015 | 26.8 ± 1.2 | 1930 | 4.76 ± 2.96 | 27.2 ±6.1 | 26.4 ±6.2 | 28.4 ±3.6 | 391.4 ± 104.7 | 0.69 ± 0.21 | 6.8 ± 7.2 |
| Reference period | 26.5 ± 1.1 [D] | 2030 | 4.0 ± 2.0 [D] | 28.3 ± 1.7 [E] | 29.9 ± 1.8 [F] | 25.5 ± 2.0 [E] | 397.6 ± 103.6 [G] | - | - |

A) no data 26.07.-06-09.2015, B) no data 05.08.-06.09.2015, C) no data 14.12.-31.12.2015, D) no data 30.08.-0.09.2014, 12.01.04.02.2016, 14.04.-11.05.2016, E) no data 31.05.-10.09.2014, 01.01.-04.02.2016, 14.04.11.05.2016, F) no data 31.05.-10.09.2014, 01.01.-11.02.2016, 14.04.-11.05.2016, G) no data 31.05.-08.09.2014, 12.01.-04.02.2016, 14.04.-11.05.2016

**Table 2: Net $CO_2$ flux, maximum rate of photosynthesis ($A_{max}$), accumulated carbon, atmospheric $CO_2$-concentration, Bowen ratio, evapotranspiration ($ET$) and actual $ET$ divided by potential $ET$ ($ET/ET_{pot}$) (daily mean ± SD, or accumulated for precipitation and carbon) derived from 30-minute averages or daily average ($A_{max}$, Bowen ratio) during pre-drought, non-haze drought, haze drought and post-haze period in 2015, for the entire year 2015 and the reference period May 2014-December 2014, January 2016-July 2016.**

| Period | Net $CO_2$ flux (net ecosystem exchange) [μmol m$^{-2}$ s$^{-1}$] | Maximum rate of photosynthesis ($A_{max}$) [μmol m$^{-2}$ s$^{-1}$] | Accumulated carbon [g C m$^{-2}$] | $CO_2$ concentration [ppm] | Bowen ratio | Evapotranspiration ($ET$) (mm d$^{-1}$) | $ET/ET_{pot}$ |
|---|---|---|---|---|---|---|---|
| Pre-drought (128 days) | -2.10 ± 12.91 | 27.4 ± 8.1 | 278.6 ± 81.8 | 416 ± 29 | 0.12 ± 0.10 | 3.6 ± 4.9 | 0.55 ± 0.11 |
| Drought (127 days) | -2.33 ± 14.07 | 26.6 ± 5.1 | 306.8 ± 91.1 | 412 ± 25 | 0.13 ± 0.13 | 3.7 ± 4.8 | 0.45 ± 0.09 |
| Haze (61 days) | -0.33 ± 12.70 | 31.4 ± 8.3 | 23.0 ± 5.5 | 429 ± 26 | 0.16 ± 0.14 | 2.5 ± 3.5 | 0.45 ± 0.07 |
| Post-haze (49 days) | -1.41 ± 14.50 | 29.1 ± 6.6 | 69.1 ± 20.0 | 429 ± 29 | 0.14 ± 0.14 | 3.4 ±4.6 | 0.48 ± 0.11 |
| 2015 | -1.79 ± 13.53 | 28.0 ± 7.2 | 676.6± 199.2 | 418 ± 28 | 0.13 ± 0.12 | 3.4 ± 4.6 | 0.49 ± 0.11 |
| Reference period | -2.20 ± 14.48 | 31.8 ± 8.4 [G] | 829.3 ± 242.3 | 407± 30 | 0.09 ± 0.05 | 4.3 ± 5.5 | 0.59 ± 0.15 |

G) no data 31.05.-08.09.2014, 12.01.-04.02.2016, 14.04.-11.05.2016