# Peer review of "El Niño–Southern Oscillation (*ENSO*) event reduces CO2 uptake of an Indonesian oil palm plantation"

_Biogeosciences, 2019_

## Referee Comment (RC1) · Anonymous Referee #1 · 18 Apr 2019

Overall, this is a well written manuscript with some interesting insights. Not only the in-situ observations, the authors also use a multi linear regression model in this study. The authors aim to investigate the impacts of ENSO events on an oil palm plantation from the aspects of CO2, water and energy exchange. The manuscript contains a clear and concise title. However, I do feel the information are overloaded in the manuscript in which the readers may find it difficult to explicitly articulate the key points. The authors also discussed the response of oil palm (NEE) to drought and haze conditions solely on the productivity aspect. It is however not clear about the relative contribution of GPP to the NEE. I find it is a bit misleading – was the ecosystem respiration also affected by drought and haze? There are in fact several publications on the effect of ENSO events

on the ecosystem productivity either in oil palm plantation, forest or other ecosystems. However, I did not see the authors discussed or compared their results with that of the published findings. It is also interesting to note that oil palm plantation was a net sink of CO2 during the ENSO year. Please find the specific comments below.

Page 2 line 17: The life cycle of oil palm is about 25 years.

I see NEE is first written on Page 3 line 2 in the manuscript, please define NEE or what does NEE stand for.

Page 3 line 26: The superscript should be put for -1 (2235 mm yr-1).

Page 3 line 25-26: I don't understand the use of climatic data from the meteorological station. If it is to show longer term data, then it is probably necessary to compare meteorological data from the site and that of the station even though they are only 29 km apart. This is to show that the longer term data is relevant to the site.

Page 3 line 30: The superscript should be put for m2 m-2.

Page 3 line 30: The LAI was very low for the palm age. Could this be due to the large gaps because of palm leaning?

Page 4 line 13: The superscript should be put for . . .m s-1.

Page 8 line 6-9: The monthly oil palm yield does not make sense as the values are extremely high. Annual fresh fruit bunch yield can rarely achieve more than 40 t/ha on average.

Page 9 line 16: The word 'NDH+ should be 'NHD+.

---

## Referee Comment (RC2) · Anonymous Referee #2 · 19 Apr 2019

Stiegler et al. study the response of surface-atmosphere fluxes from an oil palm plantation to variability in climate including fire-induced haze. The analysis is important but there are many aspects of the manuscript which should be improved before it is ready for publication.

On p 2 L 25 note that this is for the same total amount of PAR. Haze also decreases PAR at the surface and can decrease net photosynthesis for this reason.

p3 L20: this notion wasn't fully developed in the introduction. A good place to start might be: Steiner, A. L., Mermelstein, D., Cheng, S. J., Twine, T. E., & Oliphant, A. J. (2013). Observed impact of atmospheric aerosols on the surface energy budget. Earth

Interactions, 17(14), 1–22. https://doi.org/10.1175/2013EI000523.1

More detail about how transformation and adding an intercept reduced goodness-of-fit would be forthcoming. Especially adding the intercept; it is unclear to me how adding more parameters (in this case an intercept) would make goodness of fit worse. Also, are all of the terms necessary? Information criteria-type analyses (e.g. AIC, BIC) can help discriminate against unnecessary terms to come to a simpler and more robust synthesis. e.g. on L 30 p 5, all of these terms may be 'significant', but some may be relatively unimportant for explaining the variance of observations and can perhaps be safely excluded from the model.

What was the cost function for determining parameters? Least squares?

I don't really understand section 2.3.1. Is this a type of sensitivity analysis? How does this add to an already unique analysis?

3.2 and elsewhere: expressing fluxes as means of half hourly values plus or minus standard error can be misleading: do these values integrate the same proportion of daytime and nighttime data? If one of the time periods has more nighttime data due to seasonal differences in prevailing winds, the values could be different for this reason. (The paragraph beginning line 22 is better.)

bottom of p 7: define 'dim light'. Light that is 'dim' to our eyes is probably below the $CO_2$ compensation point (because human eyes respond logarithmically to light levels).

The paragraph on L 10 p 8 is unconvincing: was energy flux partitioning impacted by haze in addition to surface drying or was the latter the most important? Energy flux analyses in the manuscript could be better-developed as a whole.

Section 3.4: I'm not sure how extending the analyses behind the range of variability observed in the (linear) models is a good way to estimate the impacts of additional haze. This could bring for example far more ozone, which was not considered and is probably critical for photosynthesis here. In brief, I recommend dropping the intensified

drought/haze analysis with a non-mechanistic model and adding instead more detail about sensible and latent heat fluxes, the analysis of which at the moment seems like an afterthought.

4.1: 'relatively resistant against drying soil'...with respect to the range of drying observed here. It probably just wasn't quite dry enough rather than the plants being insensitive to soil moisture.

Good detail about oil palm physiology throughout. More biogeochemical/biogeophysical studies should include these important details about fruiting, etc.

Interesting that oil palm is insensitive to VPD up to 17 hPa given the rather large sensitivity of other tropical plants to VPD, see:

Fu, Z., et al., 2019. The surface-atmosphere exchange of carbon dioxide in tropical rainforests: Sensitivity to environmental drivers and flux measurement methodology. Agric. For. Meteorol. 263, 292-307.

Kiew, F., et al., 2018. CO2 balance of a secondary tropical peat swamp forest in Sarawak, Malaysia. Agric. For. Meteorol. 248, 494–501.

Wu, J., et al., 2017. Partitioning controls on Amazon forest photosynthesis between environmental and biotic factors at hourly to inter- annual timescales. Glob. Change Biol. 23, 1240–1257.

Section 4.2 is likewise weak...the model cannot consider the impacts of elevated temperatures beyond temperature optimums on reducing photosynthesis. Include instead perhaps an analysis of energy fluxes, which comprise hypothesis b and are never adequately described thereafter.

Conclusions and elsewhere: some discussion of ozone would be forthcoming. This isn't measured (and rarely is) but may (or may not) be important here.

---

## Author Comment (AC1) · 10 May 2019

**Response to referee comment RC1:**

*We thank the referee for reviewing our work. We place the referee's comments as "C" and provide our response in italic as "R".*

C1: Overall, this is a well written manuscript with some interesting insights. Not only the in-situ observations, the authors also use a multi linear regression model in this study. The authors aim to investigate the impacts of ENSO events on an oil palm plantation from the aspects of CO2, water and energy exchange. The manuscript contains a clear and concise title. However, I do feel the information are overloaded in the manuscript in which the readers may find it difficult to explicitly articulate the key points.

*R1: We will adapt the manuscript with a clearer storyline to ensure better readability and better articulation of the key points.*

C2: The authors also discussed the response of oil palm (NEE) to drought and haze conditions solely on the productivity aspect. It is however not clear about the relative contribution of GPP to the NEE. I find it is a bit misleading – was the ecosystem respiration also affected by drought and haze?

*R2: In an earlier stage of the data analysis we tested two approaches to partition NEE into respiration and GPP. In the first step we used the method developed by Reichstein et al., 2005 (online flux partitioning tool: http://www.bgc-jena.mpg.de/~MDIwork/eddyproc/index.php) where NEE can be modelled and separated into GPP and respiration. Although modelled and measured NEE showed a relatively high $R^2$ of 0.82, modelled NEE was on average, 58% lower as compared with measured NEE. The flux partitioning generally fails in other ecosystems as well.*

*We also tested flux partitioning of NEE into GPP and respiration using CLM-Palm (Fan et al., 2015). CLM-Palm was developed for simulating oil palm physiology, such as growth, yield, carbon, water and energy exchange. CLM-Palm is a sub-model within the framework of the Community Land Model (CLM4.5) (Oleson et al., 2013). CLM-Palm has proven capacity to accurately simulate the site-level and regional water fluxes (Meijide et al., 2017; Fan et al. 2019) as well as growth, yield and carbon fluxes (Fan et al., 2015) during non-ENSO periods. In comparison with measured NEE, CLM-model has fairly good performance in pre-drought and post-haze periods but it struggled to represent daily average NEE during the non-haze drought and haze drought periods (see Fig. 1 below). We speculate that the model is oversensitive to extreme meteorological events, such as drought and haze, due to the model's soil water stress function (Sellers et al., 1986) and missing plant hydraulic processes in the overarching CLM4.5 (Oleson et al., 2013). CLM5, which has been released recently, has implemented plant hydraulic functions which allow to simulate the utilization of trunk water storage by oil palm during dry periods but CLM-Palm has not been adapted to the CLM5 framework yet.*

[Figure]

*Figure 1. Comparison between diurnal trends of net ecosystem exchange (NEE) from eddy covariance (EC) measurements and CLM4.5-Palm model output during pre-drought, drought, haze and post-haze period. Shaded areas represent 95 % confidence limits.*

*Therefore, we decided to solely focus on NEE since our main focus in this manuscript lies on the overall $CO_2$ flux behaviour of the oil palm plantation during the extreme events of drought and haze. We also present night time ecosystem respiration and we were able to disentangle the driving parameters of night time respiration with our multiple linear regression model (MLRM). In the updated version of the manuscript we will add more information on the expected and possible behaviour of day time respiration affected by non-haze drought and haze drought.*

- *Reichstein, M., et al. (2005): On the separation of net ecosystem exchange into assimilation and ecosystem respiration: review and improved algorithm, global Change Biology 11, 1424-1439.*
- *Meijide, A. et al. (2017): Controls of water and energy fluxes in oil palm plantations: Environmental variables and oil palm age, Agricultural and Forest Meteorology 239, 71-85.*
- *Fan, Y. et al. (2015): A sub-canopy structure for simulating oil palm in the Community Land Model (CLM-Palm): phenology, allocation and yield, Geosci. Model Dev. 8, 3785-3800.*
- *Oleson, K., et al. (2013): Technical Description of version 4.5 of the Community Land Model (CLM), NCAR/TN-503+STR, NCAR Technical Note, National Center for Atmospheric Research, Boulder, Colorado, 434 p.*
- *Fan, Y. et al. (2019): Reconciling canopy interception parameterization and rainfall forcing frequency in the Community Land Model for simulating evapotranspiration of rainforests and oil palm plantations in Indonesia, Journal of Advances in Modeling Earth Systems, 11, 732-751.*
- *Sellers, P. et al. (1986): A simple biosphere model (SiB) for use within general circulation models, J. Atmos. Sci. 43 (6), 505-531.*

C3: There are in fact several publications on the effect of ENSO events on the ecosystem productivity either in oil palm plantation, forest or other ecosystems. However, I did not see the authors discussed or compared their results with that of the published findings.

*R3: We will update the manuscript with a discussion on the effect of ENSO events on the ecosystem productivity in other ecosystems such as forests and plantations.*

C4: It is also interesting to note that oil palm plantation was a net sink of CO2 during the ENSO year. Please find the specific comments below.

*R4: ENSO in 2015 was characterized by a distinct drought period which lasted in our study region from May until October 2015. During the haze drought period, the oil palm plantation was carbon neutral due to the dense smoke and overall reduction in available PAR. After the end of the haze drought period towards the beginning of the wet season we observe a short transition period where carbon uptake is relatively low compared to the rest of the wet season. However, except for the two months of the haze drought period, the oil palm plantations remained a sink of atmospheric $CO_2$. Our study site is a well-managed commercial oil palm plantation where fertilization and pest control is applied on a frequent basis. Other oil palm plantations, with less developed management practices might have lower ability to adapt to the drought and haze conditions compared to our study site.*

C5: Page 2 line 17: The life cycle of oil palm is about 25 years.

*R5: We agree with the referee. We will update the paragraph. The life cycle of oil palm is about 25 years (Woittiez et al (2017).*

- *Woittiez, L. et al. (2017): Yield gaps in oil palm: A quantitative review of contributing factors, Euop. J. Agronomy 83, 57-77.*

C6: I see NEE is first written on Page 3 line 2 in the manuscript, please define NEE or what does NEE stand for.

*R6: We will update the manuscript and define NEE (net ecosystem exchange).*

C7: Page 3 line 26: The superscript should be put for -1 (2235 mm yr-1).

*R7: We will update the paragraph.*

C8: Page 3 line 25-26: I don't understand the use of climatic data from the meteorological station. If it is to show longer term data, then it is probably necessary to compare meteorological data from the site and that of the station even though they are only 29km apart. This is to show that the longer term data is relevant to the site.

*R8: We will update the wording. The only station which has such long-term climate data available is Sultan Thaha Airport Jambi. Therefore, in the current version of the manuscript we use this data to show the overall long-term climate characteristics of the region where our meteorological tower is located. Measurements at our tower started in 2013 and we do not find any significant differences in daily average air temperature (P<0.001) or in monthly sum of precipitation (P<0.001).*

C9: Page 3 line 30: The superscript should be put for m2 m-2.

*R9: We will update the manuscript.*

C10: Page 3 line 30: The LAI was very low for the palm age. Could this be due to the large gaps because of palm leaning?

*R10: We will update the manuscript with information on how LAI at the study site was derived. In this study we did not investigate the impact of oil palm leaning on LAI. We use LAI which was estimated by Fan et al. (2015) based on the number of expanded leaves (35-45) per palm. The planting density at the site is 156 palms per ha (8x8 m horizontal density). Sample measurements of LAI (unpublished data), using LAI-2200 Plant Canopy Analyzer (LI-COR Inc. Lincoln, USA) at 12 oil palm plantation plots in the Jambi province in June 2018 show average LAI of 2.7 ±0.57 SD $m^{-2}$. Oil palm age at the measured plots is ~18 years and horizontal density varies between 8x8 m or 9x9 m. Awal & Wan Ishak (2008) report LAI of 3.05 ±0.119 $m^{-2}$ $m^{-2}$ and 4.05 ±0.343 $m^{-2}$ $m^{-2}$ for two oil palm locations with 16 years old palms and a density of 148 palms per ha. Breure (2010) reports mean LAI of 5.97-5.51 $m^{-2}$ $m^{-2}$ for three 8 years old plantations with 135 palms per ha.*

- *Fan, Y. et al. (2015): A sub-canopy structure for simulating oil palm in the Community Land Model (CLM-Palm): phenology, allocation and yield, Geosci. Model Dev. 8, 3785-3800.*
- *Awal, M.A. & Wan Ishak, W.I. (2008): Measurement of oil palm LAI by manual and LAI-2000 method, Asian Journal of Scientific Research 1 (1), 49-59.*
- *Breure, C.J. (2010): Rate of leaf expansion: A criterion for identifying oil palm (Elaeis guineensis Jacq.) types suitable for planting at high densities, NJAS – Wageningen Journal of Life Sciences 57, 141-147.*

C11: Page 4 line 13: The superscript should be put for...m s-1.

*R11: We will update the manuscript.*

C12: Page 8 line 6-9: The monthly oil palm yield does not make sense as the values are extremely high. Annual fresh fruit bunch yield can rarely achieve more than 40 t/ha on average.

*R12: We agree with the referee. We reanalysed our harvest data and found an error in the calculation of monthly yield. We apologize for the error. The wording of the paragraph will be changed with the correct monthly harvest: "From August 2015, monthly oil palm yield declined continuously from 3.93 t ha$^{-1}$ to its minimum of 1.05 t ha$^{-1}$ in May 2016. Compared to the same period (Nov.-May) in the two years before and the year after the ENSO event, average yield affected by 2015-drought and haze was 32% (0.70 t ha$^{-1}$) lower. Considering the 2015-haze drought only, average oil palm yield 6-9 months after the beginning of the haze drought was even 50% (1.1 t ha$^{-1}$) lower compared to the non-ENSO years."*

C13: Page 9 line 16: The word 'NDH+ should be 'NHD+

*R13: We will update the manuscript.*

---

## Author Comment (AC2) · 10 May 2019

**Response to referee comment RC2:**

*We thank the referee for reviewing our work. We place the referee's comments as "C" and provide our response in italic as "R".*

C1: Stiegler et al. study the response of surface-atmosphere fluxes from an oil palm plantation to variability in climate including fire-induced haze. The analysis is important but there are many aspects of the manuscript which should be improved before it is ready for publication. On p 2 L 25 note that this is for the same total amount of PAR. Haze also decreases PAR at the surface and can decrease net photosynthesis for this reason.

*R1: We agree with the referee. We will update the paragraph and add information on the impact of increasing aerosol particles from biomass burning on the total amount of PAR.*

C2: p3 L20: this notion wasn't fully developed in the introduction. A good place to start might be: Steiner, A. L., Mermelstein, D., Cheng, S. J., Twine, T. E., & Oliphant, A. J.(2013). Observed impact of atmospheric aerosols on the surface energy budget. Earth Interactions, 17(14), 1–22. https://doi.org/10.1175/2013EI000523.1

*R2: We will update the manuscript and further develop the possible impact of aerosol particles on energy flux partitioning and $CO_2$ uptake during our study period. We did not measure aerosol concentration at the study site and defined the haze period based on fraction of diffuse radiation and the persistence of high values of fraction of diffuse radiation. Steiner et al. (2013) report that increased aerosol concentration and related increase in diffuse light increase plant photosynthesis and therefore decrease the ratio of sensible to latent heat. In our study we observe an increase in the ratio of sensible to latent heat which is likely due to water stress and related partial stomata closure due to high VPD. Wang et al. (2018) observe that increased aerosol concentration increase overall canopy photosynthesis but due to different processes in sun and shaded leaves. Sun-exposed leaves benefit from lower VPD while shaded leaves benefit from increased diffuse light conditions. In our study, during the haze drought period atmospheric VPD reached its overall peak and Bowen ratio concurrently increased. In our model we also observe an overall negative impact (decrease in $CO_2$ uptake) due to the high VPD.*

*At our study site, increased fraction of diffuse radiation due to biomass burning has an overall positive impact (increase in $CO_2$ uptake) and decreased PAR a negative impact on $CO_2$ uptake, which is in line with the findings of Malavelle et al (2019). However, while Malavelle et al (2019) conclude that the positive impact of increased diffuse light conditions offsets the negative impact of decreased PAR we observe that the increase in diffuse light conditions is not able to offset the negative impact in decreased PAR. We suggest that the strong intensity and relatively long duration of the haze, with persistently high values of fraction of diffuse radiation for approx. two months, inhibits a positive impact on $CO_2$ uptake.*

- *Steiner, A.L. et al. (2013): Observed impact of atmospheric aerosols on the surface energy budget, Earth Interactions 17 (14), 1-22.*
- *Wang, X. et al. (2018): Field evidences for the positive effects of aerosols on tree growth, Global Change Biology 24, 4983-4992.*
- *Malavelle, F.F. et al. (2019): Studying the impact of biomass burning aerosol radiative and climate effects on the Amazon rainforest productivity with an Earth system model, Atmos. Chem. Phys. 19, 1301-1326.*

C3: More detail about how transformation and adding an intercept reduced goodness-of-fit would be forthcoming. Especially adding the intercept; it is unclear to me how adding more parameters (in this case an intercept) would make goodness of fit worse.

*R3: In the case of transformation, we transformed each data by subtracting the mean and dividing it by the standard deviation. The transformed data had a mean zero with a standard deviation of 1. e.g. NEE_transform = (NEE – mean(NEE))/sd(NEE) = scale(NEE). In the case of the transformed data as well as when an intercept was added in the 24-hour original NEE model, Temperature and VPD became insignificant (p-value ~ [0.5 to 0.8]), and thus the goodness of fit decreased by 53%. In what follows, we show different cases where we examined different MLRMs in relation to setting up the model:*

*Case 1: lm(formula = scale(NEE) ~ scale(VPD) + scale(CO2) + scale(fdifRad) + scale(wind) + scale(Tair))*

*Coefficients:*

| | Estimate | Std. Error | t value | Pr(>\|t\|) | |
|---|---|---|---|---|---|
| *(Intercept)* | *-0.112342* | *0.051000* | *-2.203* | *0.028650* | * |
| *scale(VPD)* | *0.020793* | *0.086572* | *0.240* | *0.810408* | |
| *scale(CO2)* | *0.177867* | *0.052516* | *3.387* | *0.000837* | *** |
| *scale(fdifRad)* | *0.121650* | *0.052972* | *2.296* | *0.022589* | * |
| *scale(wind)* | *-0.177130* | *0.053028* | *-3.340* | *0.000983* | *** |
| *scale(Tair)* | *0.009968* | *0.088540* | *0.113* | *0.910466* | |

*Case2: lm(formula = scale(NEE) ~ scale(CO2) + scale(fdifRad) + scale(wind))*

*Coefficients:*

| | Estimate | Std. Error | t value | Pr(>\|t\|) | |
|---|---|---|---|---|---|
| *(Intercept)* | *-0.10971* | *0.04848* | *-2.263* | *0.024591* | * |
| *scale(CO2)* | *0.17776* | *0.05231* | *3.399* | *0.000803* | *** |
| *scale(fdifRad)* | *0.11620* | *0.04873* | *2.385* | *0.017930* | * |
| *scale(wind)* | *-0.17344* | *0.04763* | *-3.641* | *0.000338* | *** |

*Case 3: lm(formula = NEE ~ VPD + CO2 + fdifRad + wind + Tair - 1)*
*Coefficients:*

| | Estimate | Std. Error | t value | Pr(>\|t\|) | |
|---|---|---|---|---|---|
| *VPD* | *0.126540* | *0.057204* | *2.212* | *0.02798* | * |
| *CO2* | *0.014808* | *0.008446* | *1.753* | *0.08095* | . |
| *fdifRad* | *2.144689* | *1.297013* | *1.654* | *0.09964* | . |
| *wind* | *-1.635912* | *0.288365* | *-5.673* | *4.37e-08* | *** |
| *Tair* | *-0.313711* | *0.114494* | *-2.740* | *0.00665* | ** |

*Case 4: lm(formula = NEE ~ VPD + CO2 + fdifRad + wind + Tair)*

*Coefficients:*

*Estimate Std. Error t value Pr(>|t|)*

*(Intercept) -19.78924    6.57646  -3.009 0.002926 \*\**

*VPD         0.01612    0.06711   0.240 0.810408*

*CO2         0.03960    0.01169   3.387 0.000837 \*\*\**

*fdifRad     2.99722    1.30513   2.296 0.022589 \**

*wind       -1.11112    0.33264  -3.340 0.000983 \*\*\**

*Tair        0.01772    0.15742   0.113 0.910466*

*Case 5: lm(formula = NEE ~ CO2 + fdifRad + wind)*

*Coefficients:*

*Estimate Std. Error t value Pr(>|t|)*

*(Intercept) -19.11845    4.67333  -4.091 6.01e-05 \*\*\**

*CO2         0.03958    0.01165   3.399 0.000803 \*\*\**

*fdifRad     2.86305    1.20054   2.385 0.017930 \**

*wind       -1.08794    0.29880  -3.641 0.000338 \*\*\**

| Case Number | Goodness of fit | Insignificant p-values | AIC score |
|---|---|---|---|
| 1 | 0.20 | Temperature, VPD [0.8 to 0.9] | 494 |
| 2 | 0.21 | None | 490 |
| 3 | 0.74 | None | 808 |
| 4 | 0.20 | Temperature, VPD [0.8 to 0.9] | 801 |
| 5 | 0.21 | None | 798 |

*In the above table, the AIC score differs substantially between models that used original and scaled data, where the model that used the scaled data had low values of AIC score. The model (case 3) that used the original data but excluded the intercept had a relatively high value of goodness of fit when compared with all other cases. Because the AIC score didn't change much between cases 3 and 4 and that case 3 had a relatively high goodness of fit value, we chose to use the model in case 3 for this study.*

C4: Also, are all of the terms necessary? Information criteria-type analyses (e.g. AIC, BIC) can help discriminate against unnecessary terms to come to a simpler and more robust synthesis. e.g. on L 30 p 5, all of these terms may be 'significant', but some may be relatively unimportant for explaining the variance of observations and can perhaps be safely excluded from the model.

*R4: Yes, we agree with the referee that information criteria-type analyses are important metrics that can help simplify statistical models and aid in deciding which variables to keep and which ones to discard. We did not include these metrics in our previous version of the manuscript. Now, we have included AIC scores along with the goodness of fit values for 5 different cases to show how we selected the model.*

*We would like to point out here that we initially included many more variables than specified in equations 1 to 3 in the manuscript for the model selection since we did not put a limit on the number of covariates to explain the observed NEE. In cases 1 to 5, we have showed that we did consider removing the unnecessary covariates on the basis of high p-values. Yes, we did not identify the "relatively unimportant" variables in explaining the variation in observations. If we had standardised the regression coefficients by using a "transformation approach" as we showed in some of the cases above, then we could have compared the regression coefficients to identify their relative importance; however, that was not the focus of the current study.*

C5: What was the cost function for determining parameters? Least squares?

*R5: We used three different models of NEE (as in equations 1 to 3) in the manuscript, which were the different "cost functions". We used the built in linear regression function in R ("lm") to fit the models (see cases above). Yes, the parameter estimates of the MLRMs were estimated using the ordinary least squares method.*

C6: I don't really understand section 2.3.1. Is this a type of sensitivity analysis? How does this add to an already unique analysis?

*R6: Yes, in a very general way it can be considered as a type of a sensitivity analysis. However, it is important to note that typically in a sensitivity analysis, model inputs (that are more uncertain) are varied to understand how the model outcomes change. In this case, the parameters of the model (the coefficients) would be considered more uncertain. However, we did not change the coefficients but changed the input variables (the predictors) to examine the effects on the response variable (NEE). Therefore, we would consider the analysis carried out in this section more as a prediction or a scenario type analysis rather than a sensitivity analysis – although both of them are quite closely linked.*

*This analysis helps us understand the likely impacts of changes in drought and haze on NEE.*

C7: 3.2 and elsewhere: expressing fluxes as means of half hourly values plus or minus standard error can be misleading: do these values integrate the same proportion of daytime and nighttime data? If one of the time periods has more nighttime data due to seasonal differences in prevailing winds, the values could be different for this reason. (The paragraph beginning line 22 is better.)

*R7: Due to the proximity of our study site to the equator the difference in day length between summer solstice and winter solstice is only 12 minutes. Therefore, we consider the impact of differences in day length on the fluxes as negligible. Average friction velocity (u\*) during both day and night time, is slightly higher during the pre-drought period ($0.26 \pm 0.16\ m\ s^{-1}$) as compared to the other periods ($0.21 \pm 0.14\ m\ s^{-1}$).*

C8: bottom of p 7: define 'dim light'. Light that is 'dim' to our eyes is probably below the CO2 compensation point (because human eyes respond logarithmically to light levels).

*R8: We will update the manuscript and change the wording. In this specific case, dim light conditions refer to a reduction in the overall day time (6:00-18:30 h local time) levels of PAR and incoming solar radiation during the haze period and not, as the current wording might suggest, to dim light conditions during dusk or dawn.*

C9: The paragraph on L 10 p 8 is unconvincing: was energy flux partitioning impacted by haze in addition to surface drying or was the latter the most important? Energy flux analyses in the manuscript could be better-developed as a whole.

*R9: We will update the paragraph. Soil moisture continued to decrease over the non-haze drought and haze drought period (Table 1) but the overall decrease in deep layer (100 cm) soil moisture was less pronounced as in 30 cm and 60 cm depth. Oil palm is able to uptake water from deep soil and store the water in the trunk during night that supports water use during peak hours of photosynthesis (Niu et al., 2015; Meijide et al., 2017). Therefore, soil moisture might only be a minor factor of the observed changes in energy flux partitioning. We will add more information on energy flux partitioning into sensible and latent heat in the manuscript.*

- *Niu, F. et al. (2015): Oil palm water use: calibration of a sap flux method and a field measurement scheme, Tree Physiol. 35 (5), 563-573.*
- *Meijide, A. et al. (2017): Controls of water and energy fluxes in oil palm plantations: Environmental variables and oil palm age, Agricultural and Forest Meteorology 239, 71-85.*

C10: Section 3.4: I'm not sure how extending the analyses behind the range of variability observed in the (linear) models is a good way to estimate the impacts of additional haze. This could bring for example far more ozone, which was not considered and is probably critical for photosynthesis here. In brief, I recommend dropping the intensified drought/haze analysis with a non-mechanistic model and adding instead more detail about sensible and latent heat fluxes, the analysis of which at the moment seems like an afterthought.

*R10: We thank the referee for raising the concern whether extending the analyses behind the range of variability observed in the data is a good way to estimate the impacts of additional haze. This is an important point that the referee raised. Indeed, this is a limitation for not only statistical models but also for mechanistic models, where both of the models may not realistically estimate the impacts of additional haze unless they are developed using that range in the first place. However, numerous mechanistic land surface models have been run on domains where they looked at responses of these models to future climatic conditions and also at large time-steps such as on century time-scales. The predictions of such models can involve relatively large uncertainties. On the grounds that we did not consider relatively large changes in the variables (i.e. we only considered +/- 20%) and the time-step that we think it might occur is not more than 5 years. Daily average NEE during the haze drought period ranged between -3.61 and 4.80 $\mu mol\ m^{-2}\ s^{-1}$.*

*We will make this statement clear in the manuscript and so we acknowledge that the outcomes of our model application has some limitations and is simple but we think it can still be useful, for e.g. it can serve as a hypothesis that can be looked into in the future as more data becomes available.*

C11: 4.1: 'relatively resistant against drying soil'...with respect to the range of drying observed here. It probably just wasn't quite dry enough rather than the plants being insensitive to soil moisture.

*R11: We agree with the referee and we will update the manuscript and change the paragraph. Oil palm is able to uptake water from deep soil and store the water in the trunk during night that supports water use during peak hours of photosynthesis (Niu et al., 2015; Meijide et al., 2017). Soil moisture conditions in the deeper soil layer (100 cm depth) showed a relatively moderate decrease during both non-haze drought and haze drought period and remained higher as compared to soil moisture conditions in the upper layers (30 cm and 60 cm depth) (Table 1). Therefore, with respect to the range of drying soil observed in this study, the relatively moderate decrease in soil moisture in deeper soil layers was not reflected in a decrease in NEE.*

- *Niu, F. et al. (2015): Oil palm water use: calibration of a sap flux method and a field measurement scheme, Tree Physiol. 35 (5), 563-573.*
- *Meijide, A. et al. (2017): Controls of water and energy fluxes in oil palm plantations: Environmental variables and oil palm age, Agricultural and Forest Meteorology 239, 71-85.*

C12: Good detail about oil palm physiology throughout. More biogeochemical/biogeophysical studies should include these important details about fruiting, etc.

*R12: We will update the manuscript with more information on fruiting and other details about oil palm physiology.*

C13: Interesting that oil palm is insensitive to VPD up to 17 hPa given the rather large sensitivity of other tropical plants to VPD, see: Fu, Z., et al., 2019. The surface-atmosphere exchange of carbon dioxide in tropical rainforests: Sensitivity to environmental drivers and flux measurement methodology. Agric. For. Meteorol. 263, 292-307. Kiew, F., et al., 2018. $CO_2$ balance of a secondary tropical peat swamp forest in Sarawak, Malaysia. Agric. For. Meteorol. 248, 494–501. Wu, J., et al., 2017. Partitioning controls on Amazon forest photosynthesis between environmental and biotic factors at hourly to inter- annual timescales. Glob. ChangeBiol. 23, 1240–1257.

*R13: We will update the manuscript with a discussion on the effect of VPD on oil palm compared with other tropical plants as suggested by the referee.*

C14: Section 4.2 is likewise weak...the model cannot consider the impacts of elevated temperatures beyond temperature optimums on reducing photosynthesis. Include instead perhaps an analysis of energy fluxes, which comprise hypothesis b and are never adequately described thereafter.

*R14: We also agree with the referee here that the current model is built on the dataset that may not have included elevated temperatures which might be clearly important for downregulating oil-palm photosynthesis. Indeed, we do have data-sets covering a few more years which show that air temperature is within the range of the 2015-ENSO year. Our current data does therefore include elevated temperatures to an extent and so we are focusing on short-term responses.*

*Rising $CO_2$ and the deforestation of surrounding forests can likely enhance temperatures of oil-palm in the future. In either of these cases, our model application might not be suitable. Therefore, in the updated version of the manuscript we will change the term "future" into "short-term response of oil palm to changed climatic conditions". These short-term responses of oil palm focus on the current life cycle of the oil palm plantation, which was planted in 2002 and is therefore closer to rotation now, which happens 25 years after the planting. Therefore, these short-term responses do not include elevated temperature*

*associated with rising $CO_2$ levels beyond the temperature optimum of oil palm photosynthesis.*

C15: Conclusions and elsewhere: some discussion of ozone would be forthcoming. This isn't measured (and rarely is) but may (or may not) be important here.

*R15: We will update the manuscript with a discussion on possible impact of ozone on oil palm photosynthesis. Ground-level ozone exerts strong toxicity on tropical and sub-tropical agricultural and natural vegetation (Moraes et al., 2004; Felzer et al., 2007; Zhang et al., 2014; Chen et al., 2018). Ozone concentration has not been measured in this study but biomass burning is considered to significantly affect near-surface ozone concentration due to emission of ozone precursor gases (Kita et al., 2000) and fire air pollution generally leads to a decrease in gross primary productivity (GPP) (Yue & Unger, 2010). To our knowledge, no study has focused on ozone concentration from biomass burning during the 2015 ENSO event but studies observe a strong increase in ozone concentration from biomass burning during the 1997-ENSO (Thompson et al., 2001) and during the 2006-ENSO event (Nassar et al., 2009). At our study site, we therefore expect an increase in ground-level ozone concentration during the haze drought period which might have negatively affected oil palm carbon sequestration.*

- *Moraes, R.M. et al. (2004): Photosynthetic responses of tropical trees to short-term exposure to ozone, Photosynthetica 42 (2), 291-293.*
- *Felzer, B.S. et al. (2007): Impacts of ozone on trees and crops, C. R. Geoscience 339, 784-798.*
- *Zhang, W. et al. (2014): Elevated ozone negatively affects photosynthesis of current-year leaves but not previous-year levae in evergreen Cyclobalanopsis glauca seedlings, Environmental Pollution 184, 676-681.*
- *Chen, Z. et al. (2018): Effects of elevated ozone levels on photosynthesis, biomass and non-structural carbohydrates of Phoebe bournei and Phoebe zhennan in subtropical China, Front. Plant Sci. 9: 1764.*
- *Kita, K. et al. (2000): Total ozone increase associated with forest fires over the Indonesian region and its relation to the El Niño-Southern oscillation, Atmospheric Environment 34, 2681-2690.*
- *Yue, X. & Unger, N. (2010): Fire air pollution reduces global terrestrial productivity, Nature Communications 9: 5413.*
- *Thompson, A.M. et al. (2001): Tropical tropospheric ozone and biomass burning, Science 291, 2128-2132.*
- *Nassar, R. et al. (2009): Analysis of tropical tropospheric ozone, carbon monoxide, and water vapour during the 2006 El Niño using TES observations and the GEOS-Chem model, J. Geophys. Res. 114, D17304.*

---

## Author Response (AR1)

**Response to referee comment RC1:**

*We thank the referee for reviewing our work. We place the referee's comments as "C" and provide our response in italic as "R".*

C1: Overall, this is a well written manuscript with some interesting insights. Not only the in-situ observations, the authors also use a multi linear regression model in this study. The authors aim to investigate the impacts of ENSO events on an oil palm plantation from the aspects of CO2, water and energy exchange. The manuscript contains a clear and concise title. However, I do feel the information are overloaded in the manuscript in which the readers may find it difficult to explicitly articulate the key points.

*R1: We adapted the manuscript with a clearer storyline in the results (3.3) and discussion section (4.1, 4.2) to ensure better readability and better articulation of the key points.*

C2: The authors also discussed the response of oil palm (NEE) to drought and haze conditions solely on the productivity aspect. It is however not clear about the relative contribution of GPP to the NEE. I find it is a bit misleading – was the ecosystem respiration also affected by drought and haze?

*R2: We updated the methods section with the following paragraph:*

*Initially, we applied $CO_2$ flux partitioning of NEE into gross primary production (GPP) and respiration using (a) a non-linear regression model based on Reichstein et al. (2005) and (b) $CO_2$ flux partitioning based on CLM-Palm (Fan et al., 2015) which is a sub-model within the framework of the Community Land Model (CLM4.5) (Oleson et al., 2013). The non-linear regression model underestimated NEE by 58%, on average, most likely because the model struggles to assess the temperature sensitivity of ecosystem respiration using the filtered night time data (Oikawa et al., 2017). CLM-Palm struggled to represent daily average NEE during the non-haze drought and haze drought periods, most likely due to the models' soil water stress function (Sellers et al., 2013) and missing plant hydraulic processes in the overarching CLM4.5 (Oleson et al., 2013). Therefore, we decided to solely focus on NEE to describe the overall $CO_2$ flux behaviour of the oil palm plantation during the extreme events of drought and haze. However, we used the night time NEE (=respiration) as a proxy for the overall behaviour of oil palm monoculture respiration and disentangled its driving climatic variables.*

*Further, we updated the manuscript with a discussion on the behaviour of respiration:*

*Temperature increase and related heat stress is another factor which might negatively affect the growth of oil palm (Oettli et al., 2018). Our analysis did not support this finding because during the non-haze drought the effect of increasing temperature on NEE was almost negligible. During the haze drought, higher air temperature had a positive impact on CO2 uptake although the haze period experienced the highest air temperature during the entire study period. Changes in temperature and moisture availability also impact oil palm ecosystem respiration. Matysek et al. (2018) observed high heterotrophic carbon loss from drained peat soils in a Malaysian oil palm plantation during the dry season and Sigau & Hamid (2018) found similar behaviour in Malaysian rubber and oil palm plantations on drying Haplic Nitisols soils but both studies report only minor impact of increased soil temperature on soil respiration. Autotrophic respiration, however, tends to decrease with increasing leaf temperature (Slot et al., 2014). In our study, the increase in air temperature tended to increase night time ecosystem respiration and therefore might also lead to higher day time respiration during the non-haze drought and haze drought period.*

C3: There are in fact several publications on the effect of ENSO events on the ecosystem productivity either in oil palm plantation, forest or other ecosystems. However, I did not see the authors discussed or compared their results with that of the published findings.

*R3: We updated the manuscript with a discussion on the effect of ENSO events on the ecosystem productivity in other ecosystems such as forests and plantations (section 4.1).*

C4: It is also interesting to note that oil palm plantation was a net sink of $CO_2$ during the ENSO year. Please find the specific comments below.

*R4: ENSO in 2015 was characterized by a distinct drought period which lasted in our study region from May until October 2015. During the haze drought period, the oil palm plantation was carbon neutral due to the dense smoke and overall reduction in available PAR. After the end of the haze drought period towards the beginning of the wet season we observe a short transition period where carbon uptake is relatively low compared to the rest of the wet season. However, except for the two months of the haze drought period, the oil palm plantations remained a sink of atmospheric $CO_2$. Our study site is a well-managed commercial oil palm plantation where fertilization and pest control is applied on a frequent basis. Other oil palm plantations, with less developed management practices might have lower ability to adapt to the drought and haze conditions compared to our study site.*

C5: Page 2 line 17: The life cycle of oil palm is about 25 years.

*R5: We agree with the referee. We updated the wording according to the referees comment: "Oil palm has high life cycle of about 25 years (Woittiez et al., 2017)…"*

C6: I see NEE is first written on Page 3 line 2 in the manuscript, please define NEE or what does NEE stand for.

*R6: We updated the manuscript and defined NEE (net ecosystem exchange) in its first appearance in the manuscript. NEE was defined as GPP (gross primary productivity) plus respiration. In this study we assign fluxes as positive when they are directed away from the surface.*

C7: Page 3 line 26: The superscript should be put for -1 (2235 mm yr-1).

*R7: We updated the paragraph.*

C8: Page 3 line 25-26: I don't understand the use of climatic data from the meteorological station. If it is to show longer term data, then it is probably necessary to compare meteorological data from the site and that of the station even though they are only 29km apart. This is to show that the longer term data is relevant to the site.

*R8: We updated the paragraph with the following sentences:*

*"A comparison of air temperature and precipitation at our study site with climate records from Sultan Thaha Airport Jambi during our study period June 2014 to July 2016 showed no significant differences in daily average air temperature ($P<0.001$) or in monthly accumulated precipitation ($P<0.001$). Therefore, we consider the long-term climate records being representative for our study location."*

C9: Page 3 line 30: The superscript should be put for m2 m-2.

*R9: We updated the manuscript.*

C10: Page 3 line 30: The LAI was very low for the palm age. Could this be due to the large gaps because of palm leaning?

*R10: We updated the paragraph with information on how LAI at the study site was derived. Palms are planted in a triangular array, with 8x8 m horizontal density and 156 palms per ha. Based on this horizontal density, an average palm height of 12 m, and 35-45 expanded leaves per palm, Fan et al. (2015) estimated a site-specific leaf area index (LAI) of 3.64 $m^2$ $m^{-2}$. Gaps in oil palms that can be created due to disturbances or extreme weather conditions was not observed in this study.*

*Sample measurements of LAI (unpublished data), using LAI-2200 Plant Canopy Analyzer (LI-COR Inc. Lincoln, USA) at 12 oil palm plantation plots in the Jambi province in June 2018 show average LAI of 2.7 ±0.57 SD $m^{-2}$. Oil palm age at the measured plots is ~18 years and horizontal density varies between 8x8 m or 9x9 m. Awal & Wan Ishak (2008) report LAI of 3.05 ±0.119 $m^{-2}$ $m^{-2}$ and 4.05 ±0.343 $m^{-2}$ $m^{-2}$ for two oil palm locations with 16 years old palms and a density of 148 palms per ha. Breure (2010) reports mean LAI of 5.97-5.51 $m^{-2}$ $m^{-2}$ for three 8 years old plantations with 135 palms per ha.*

- *Fan, Y. et al. (2015): A sub-canopy structure for simulating oil palm in the Community Land Model (CLM-Palm): phenology, allocation and yield, Geosci. Model Dev. 8, 3785-3800.*
- *Awal, M.A. & Wan Ishak, W.I. (2008): Measurement of oil palm LAI by manual and LAI-2000 method, Asian Journal of Scientific Research 1 (1), 49-59.*
- *Breure, C.J. (2010): Rate of leaf expansion: A criterion for identifying oil palm (Elaeis guineensis Jacq.) types suitable for planting at high densities, NJAS – Wageningen Journal of Life Sciences 57, 141-147.*

C11: Page 4 line 13: The superscript should be put for...m s-1.

*R11: We updated the manuscript.*

C12: Page 8 line 6-9: The monthly oil palm yield does not make sense as the values are extremely high. Annual fresh fruit bunch yield can rarely achieve more than 40 t/ha on average.

*R12: We agree with the referee. We reanalysed our harvest data and found an error in the calculation of monthly yield. We apologize for the error. We changed the wording with the correct monthly harvest:*

*"From August 2015, monthly oil palm yield declined continuously from 3.93 t $ha^{-1}$ to its minimum of 1.05 t $ha^{-1}$ in May 2016. Compared to the same period (Nov.-May) in the two years before and the year after the ENSO event, average yield affected by 2015-drought and haze was 32% (0.70 t $ha^{-1}$) lower. Considering the 2015-haze drought only, average oil palm yield 6-9 months after the beginning of the haze drought was even 50% (1.1 t $ha^{-1}$) lower compared to the non-ENSO years."*

C13: Page 9 line 16: The word 'NDH+ should be 'NHD+

*R13: We updated the manuscript.*

**Response to referee comment RC2:**

*We thank the referee for reviewing our work. We place the referee's comments as "C" and provide our response in italic as "R".*

C1: Stiegler et al. study the response of surface-atmosphere fluxes from an oil palm plantation to variability in climate including fire-induced haze. The analysis is important but there are many aspects of the manuscript which should be improved before it is ready for publication. On p 2 L 25 note that this is for the same total amount of PAR. Haze also decreases PAR at the surface and can decrease net photosynthesis for this reason.

*R1: We agree with the referee. We updated the paragraph according to the referees comment.*

C2: p3 L20: this notion wasn't fully developed in the introduction. A good place to start might be: Steiner, A. L., Mermelstein, D., Cheng, S. J., Twine, T. E., & Oliphant, A. J.(2013). Observed impact of atmospheric aerosols on the surface energy budget. Earth Interactions, 17(14), 1–22. https://doi.org/10.1175/2013EI000523.1

*R2: We updated the manuscript and further developed the possible impact of aerosol particles on energy flux partitioning and $CO_2$ uptake during our study period. We did not measure aerosol concentration at the study site and defined the haze period based on fraction of diffuse radiation and the persistence of high values of fraction of diffuse radiation.*

*Increased aerosol concentration from biomass burning and related increase in diffuse light increase plant photosynthesis and therefore decrease the ratio of sensible to latent heat (Steiner et al., 2013). However, in our study and during the peak of the drought when forest fires started to develop in the area, we observed increase in the ratio of sensible to latent heat (Bowen ratio) which is likely due to water stress and related partial stomata closure at high VPD (Dufrene & Saugier, 1993; Oettli et al., 2018). Further, increased aerosol concentration is able to increase overall canopy photosynthesis under moderately enhanced diffuse light conditions (Knohl et al., 2008; Mercado et al., 2009; Kanniah et al., 2012) and sun-exposed leaves seem to benefit from lower VPD while shaded leaves benefit from increased diffuse light conditions (Wang et al., 2018). Although our measurements and MLRM suggest that the leaves benefitted from the increase in diffuse light conditions during the haze drought period, the high level of VPD, especially during midday, was an overall stress factor for the oil palm plantation and therefore resulted in a decrease in $CO_2$ uptake. At our study site, increased fraction of diffuse radiation due to biomass burning had an overall positive impact (increase in $CO_2$ uptake) and decreased incoming PAR a negative impact on $CO_2$ uptake, which is in line with the findings of Malavelle et al (2019). However, while the authors of that study conclude that the positive impact of increased diffuse light conditions offsets the negative impact of decreased PAR we observe that the increase in diffuse light conditions is not able to offset the negative impact in decreased PAR. We suggest that the strong intensity and relatively long duration of the haze, with persistently high values of fraction of diffuse radiation for approx. two months, exceeded an optimal range of diffuse fraction (Knohl et al., 2008) and therefore inhibited a positive impact on $CO_2$ uptake.*

C3: More detail about how transformation and adding an intercept reduced goodness-of-fit would be forthcoming. Especially adding the intercept; it is unclear to me how adding more parameters (in this case an intercept) would make goodness of fit worse.

*R3: We updated the methods section (2.3 & 2.3.1) and the supplementary material (Table S2) accordingly to the referees' comment.*

*In the case of transformation, we transformed each data by subtracting the mean and dividing it by the standard deviation. The transformed data had a mean zero with a standard deviation of 1. e.g. NEE_transform = (NEE – mean(NEE))/sd(NEE) = scale(NEE). In the case of the transformed data as well as when an intercept was added in the 24-hour original NEE model, Temperature and VPD became insignificant (p-value ~ [0.5 to 0.8]), and thus the goodness of fit decreased by 53%. In what follows, we show different cases where we examined different MLRMs in relation to setting up the model:*

*Case 1: lm(formula = scale(NEE) ~ scale(VPD) + scale(CO2) + scale(fdifRad) +*

  *scale(wind) + scale(Tair))*

*Coefficients:*

|  | Estimate | Std. Error | t value | Pr(>\|t\|) |
|---|---|---|---|---|
| *(Intercept)* | *-0.112342* | *0.051000* | *-2.203* | *0.028650 \** |
| *scale(VPD)* | *0.020793* | *0.086572* | *0.240* | *0.810408* |
| *scale(CO2)* | *0.177867* | *0.052516* | *3.387* | *0.000837 \*\*\** |
| *scale(fdifRad)* | *0.121650* | *0.052972* | *2.296* | *0.022589 \** |
| *scale(wind)* | *-0.177130* | *0.053028* | *-3.340* | *0.000983 \*\*\** |
| *scale(Tair)* | *0.009968* | *0.088540* | *0.113* | *0.910466* |

*Case2: lm(formula = scale(NEE) ~ scale(CO2) + scale(fdifRad) + scale(wind))*

*Coefficients:*

|  | Estimate | Std. Error | t value | Pr(>\|t\|) |
|---|---|---|---|---|
| *(Intercept)* | *-0.10971* | *0.04848* | *-2.263* | *0.024591 \** |
| *scale(CO2)* | *0.17776* | *0.05231* | *3.399* | *0.000803 \*\*\** |
| *scale(fdifRad)* | *0.11620* | *0.04873* | *2.385* | *0.017930 \** |
| *scale(wind)* | *-0.17344* | *0.04763* | *-3.641* | *0.000338 \*\*\** |

*Case 3: lm(formula = NEE ~ VPD + CO2 + fdifRad + wind + Tair - 1)*

*Coefficients:*

*Estimate Std. Error t value Pr(>|t|)*

*VPD      0.126540   0.057204   2.212  0.02798 \**

*CO2      0.014808   0.008446   1.753  0.08095 .*

*fdifRad  2.144689   1.297013   1.654  0.09964 .*

*wind    -1.635912   0.288365  -5.673 4.37e-08 \*\*\**

*Tair    -0.313711   0.114494  -2.740  0.00665 \*\**

*Case 4: lm(formula = NEE ~ VPD + CO2 + fdifRad + wind + Tair)*

*Coefficients:*

*Estimate Std. Error t value Pr(>|t|)*

*(Intercept) -19.78924    6.57646  -3.009 0.002926 \*\**

*VPD          0.01612    0.06711   0.240 0.810408*

*CO2          0.03960    0.01169   3.387 0.000837 \*\*\**

*fdifRad      2.99722    1.30513   2.296 0.022589 \**

*wind        -1.11112    0.33264  -3.340 0.000983 \*\*\**

*Tair         0.01772    0.15742   0.113 0.910466*

*Case 5: lm(formula = NEE ~ CO2 + fdifRad + wind)*

*Coefficients:*

*Estimate Std. Error t value Pr(>|t|)*

*(Intercept) -19.11845    4.67333  -4.091 6.01e-05 \*\*\**

*CO2          0.03958    0.01165   3.399 0.000803 \*\*\**

*fdifRad      2.86305    1.20054   2.385 0.017930 \**

*wind        -1.08794    0.29880  -3.641 0.000338 \*\*\**

| Case Number | Goodness of fit | Insignificant p-values | AIC score |
|---|---|---|---|
| 1 | 0.20 | Temperature, VPD [0.8 to 0.9] | 494 |
| 2 | 0.21 | None | 490 |
| 3 | 0.74 | None | 808 |
| 4 | 0.20 | Temperature, VPD [0.8 to 0.9] | 801 |
| 5 | 0.21 | None | 798 |

*In the above table, the AIC score differs substantially between models that used original and scaled data, where the model that used the scaled data had low values of AIC score. The model (case 3) that used the original data but excluded the intercept had a relatively high value of goodness of fit when compared with all other cases. Because the AIC score didn't change much between cases 3 and 4 and that case 3 had a relatively high goodness of fit value, we chose to use the model in case 3 for this study.*

C4: Also, are all of the terms necessary? Information criteria-type analyses (e.g. AIC, BIC) can help discriminate against unnecessary terms to come to a simpler and more robust synthesis. e.g. on L 30 p 5, all of these terms may be 'significant', but some may be relatively unimportant for explaining the variance of observations and can perhaps be safely excluded from the model.

*R4: We updated the methods section and the supplementary materials to motivate our choice of model design.*

*Yes, we agree with the referee that information criteria-type analyses are important metrics that can help simplify statistical models and aid in deciding which variables to keep and which ones to discard. We did not include these metrics in our previous version of the manuscript. Now, we have included AIC scores along with the goodness of fit values for 5 different cases to show how we selected the model.*

*We would like to point out here that we initially included many more variables than specified in equations 1 to 3 in the manuscript for the model selection since we did not put a limit on the number of covariates to explain the observed NEE. In cases 1 to 5, we have showed that we did consider removing the unnecessary covariates on the basis of high p-values. Yes, we did not identify the "relatively unimportant" variables in explaining the variation in observations. If we had standardised the regression coefficients by using a "transformation approach" as we showed in some of the cases above, then we could have compared the regression coefficients to identify their relative importance; however, that was not the focus of the current study.*

C5: What was the cost function for determining parameters? Least squares?

*R5: We used three different models of NEE (as in equations 1 to 3) in the manuscript, which were the different "cost functions". We used the built in linear regression function in R ("lm") to fit the models (see cases above). Yes, the parameter estimates of the MLRMs were estimated using the ordinary least squares method and we updated the manuscript with this information.*

C6: I don't really understand section 2.3.1. Is this a type of sensitivity analysis? How does this add to an already unique analysis?

*R6: We updated the paragraph in the manuscript.*

*Yes, in a very general way it can be considered as a type of a sensitivity analysis. However, it is important to note that typically in a sensitivity analysis, model inputs (that are more uncertain) are varied to understand how the model outcomes change. In this case, the parameters of the model (the coefficients) would be considered more uncertain. However, we did not change the coefficients but changed the input variables (the predictors) to examine the effects on the response variable (NEE). Therefore, we would consider the analysis carried out in this section more as a prediction or a scenario type analysis rather than a sensitivity analysis – although both of them are quite closely linked.*

*This analysis helps us understand the likely impacts of changes in drought and haze on NEE.*

C7: 3.2 and elsewhere: expressing fluxes as means of half hourly values plus or minus standard error can be misleading: do these values integrate the same proportion of daytime and nighttime data? If one of the time periods has more nighttime data due to seasonal differences in prevailing winds, the values could be different for this reason. (The paragraph beginning line 22 is better.)

*R7: We updated the manuscript in the methods section (2.4). Due to the proximity of our study site to the equator the difference in day length between summer solstice and winter solstice is only 12 minutes. Therefore, we consider the impact of differences in day length on the fluxes as negligible. Seasonal differences in u\*, especially during night time, might impact the performance of eddy covariance gap filling. However, we found no significant differences ($P < 0.05$) in u\* which could have affected the proportion of available night time data during the different meteorological periods. Therefore, we consider the applied gap filling procedure and derived flux averaging as robust and representative for the studied time periods.*

C8: bottom of p 7: define 'dim light'. Light that is 'dim' to our eyes is probably below the CO2 compensation point (because human eyes respond logarithmically to light levels).

*R8: We updated the paragraph and changed the wording:*

*"With the continuous development of haze in September 2015 and related absence of direct sunlight the oil palm plantation seemed to compensate for the overall haze-related reduction in incoming PAR, with a jump of Amax by 13 $\mu mol\ m^{-2}\ s^{-1}$ (37%) within a couple of days (Figure 4).*

C9: The paragraph on L 10 p 8 is unconvincing: was energy flux partitioning impacted by haze in addition to surface drying or was the latter the most important? Energy flux analyses in the manuscript could be better-developed as a whole.

*R9: We restructured the results section and added a new sub-section with the title "Evapotranspiration and turbulent heat fluxes". In this sub-section we present energy flux analyses in more detail:*

*Total evapotranspiration (ET) derived from eddy covariance (EC) latent heat flux (LE) measurements was $1245 \pm 362\ mm\ yr^{-1}$ in 2015 and $1580 \pm 469\ mm\ yr^{-1}$ during the reference period (Table 2), with a higher share of ET on precipitation during the reference period (77.9%) compared to 2015 (64.5%). During the non-haze drought and haze drought periods, the oil palm plantation experienced strong water loss from ET as ET was 2.5 and*

*1.2 times the amount of precipitation, respectively. ET was lowest during the haze drought period (Figure 5, Table 2), mainly driven by the reduction in incoming solar radiation and PAR as well as by oil palm drought and heat stress which may have triggered partial stomata closure, especially in the beginning of the non-haze drought when VPD was high (Figure 2). Conversely, partial stomata closure during high VPD as well as the absence of precipitation and related drying of the upper soil generally increased sensible heat fluxes (H) at the cost of LE and ET, reflected in the behaviour of the Bowen ratio (H/LE) (Figure 5). From the first half of the pre-drought period into the second half of the non-haze period, the Bowen ratio showed a steady but relatively small decline. However, the end of the non-haze drought and the beginning of the haze drought period mark a strong transition in the behaviour of the Bowen ratio, manifested by a strong jump, peak values of ~0.38 and average of 0.25 for approx. one month. This jump in the Bowen ratio might be related to the increasing density of the haze and related reduction in incoming PAR in combination with high VPD which decrease LE mainly via oil palm water and light stress to a greater extent than the general drying of the soil and lack of precipitation.*

C10: Section 3.4: I'm not sure how extending the analyses behind the range of variability observed in the (linear) models is a good way to estimate the impacts of additional haze. This could bring for example far more ozone, which was not considered and is probably critical for photosynthesis here. In brief, I recommend dropping the intensified drought/haze analysis with a non-mechanistic model and adding instead more detail about sensible and latent heat fluxes, the analysis of which at the moment seems like an afterthought.

*R10: We thank the referee for raising the concern whether extending the analyses behind the range of variability observed in the data is a good way to estimate the impacts of additional haze. This is an important point that the referee raised. Indeed, this is a limitation for not only statistical models but also for mechanistic models, where both of the models may not realistically estimate the impacts of additional haze unless they are developed using that range in the first place. However, numerous mechanistic land surface models have been run on domains where they looked at responses of these models to future climatic conditions and also at large time-steps such as on century time-scales. The predictions of such models can involve relatively large uncertainties. On the grounds that we did not consider relatively large changes in the variables (i.e. we only considered +/- 20%) and the time-step that we think it might occur is not more than 5 years. Daily average NEE during the haze drought period ranged between -3.61 and 4.80 $\mu$mol m$^{-2}$ s$^{-1}$.*

*We made this statement clear in the methods section of the manuscript (2.3.1) and we acknowledge that the outcomes of our model application has some limitations and is simple but we think it can still be useful, for e.g. it can serve as a hypothesis that can be looked into in the future as more data becomes available.*

C11: 4.1: 'relatively resistant against drying soil'...with respect to the range of drying observed here. It probably just wasn't quite dry enough rather than the plants being insensitive to soil moisture.

*R11: We agree with the referee. We updated the manuscript and changed the paragraph. Oil palm is able to uptake water from deep soil and store the water in the trunk during night that supports water use during peak hours of photosynthesis (Niu et al., 2015; Meijide et al., 2017). Soil moisture conditions in the deeper soil layer (100 cm depth) showed a relatively moderate decrease during both non-haze drought and haze drought period and remained higher as compared to soil moisture conditions in the upper layers (30 cm and 60 cm depth) (Table 1). Therefore, with respect to the range of drying soil observed in this study, the relatively moderate decrease in soil moisture in deeper soil layers was not reflected in a decrease in NEE.*

C12: Interesting that oil palm is insensitive to VPD up to 17 hPa given the rather large sensitivity of other tropical plants to VPD, see: Fu, Z., et al., 2019. The surface-atmosphere exchange of carbon dioxide in tropical rainforests: Sensitivity to environmental drivers and flux measurement methodology. Agric. For. Meteorol. 263, 292-307. Kiew, F., et al., 2018. CO2 balance of a secondary tropical peat swamp forest in Sarawak, Malaysia. Agric. For. Meteorol. 248, 494–501. Wu, J., et al., 2017. Partitioning controls on Amazon forest photosynthesis between environmental and biotic factors at hourly to inter- annual timescales. Glob. ChangeBiol. 23, 1240–1257.

*R12: We updated the manuscript with a discussion on the effect of elevated air temperature and VPD on oil palm compared with other tropical plants in section 4.1 as suggested by the referee.*

C13: Section 4.2 is likewise weak...the model cannot consider the impacts of elevated temperatures beyond temperature optimums on reducing photosynthesis. Include instead perhaps an analysis of energy fluxes, which comprise hypothesis b and are never adequately described thereafter.

*R13: We also agree with the referee here that the current model is built on the dataset that may not have included elevated temperatures which might be clearly important for downregulating oil-palm photosynthesis. Indeed, we do have data-sets covering a few more years which show that air temperature is within the range of the 2015-ENSO year. Our current data does therefore include elevated temperatures to an extent and so we are focusing on short-term responses.*

*Rising $CO_2$ and the deforestation of surrounding forests can likely enhance temperatures of oil-palm in the future. In either of these cases, our model application might not be suitable. Therefore, in the updated version of the manuscript we changed the term "future" into "short-term response of oil palm to changed climatic conditions". These short-term responses of oil palm focus on the current life cycle of the oil palm plantation, which was planted in 2002 and is therefore closer to rotation now, which happens 25 years after the planting. Therefore, these short-term responses do not include elevated temperature associated with rising $CO_2$ levels beyond the temperature optimum of oil palm photosynthesis. We updated the methods section (2.3.1) and the discussion section (4.2) accordingly.*

C14: Conclusions and elsewhere: some discussion of ozone would be forthcoming. This isn't measured (and rarely is) but may (or may not) be important here.

*R14: We updated the manuscript with a discussion on possible impact of increased ozone and aerosol concentration on oil palm photosynthesis.*

[revised manuscript text omitted]

**Case 1:** *lm(formula = scale(NEE) ~ scale(VPD) + scale(CO2) + scale(fdifRad) + scale(wind) + scale(Tair))*

| Coefficients: | Estimate | Std. Error | t-value | Pr (>|t|) |
|---|---|---|---|---|
| (Intercept) | -0.112342 | 0.05100 | -2.203 | 0.028650 * |
| scale(VPD) | 0.020793 | 0.086572 | 0.240 | 0.810408 |
| scale(CO2) | 0.177867 | 0.052516 | 3.387 | 0.000837 *** |
| scale(fdifRad) | 0.121650 | 0.052972 | 2.296 | 0.022589 * |
| scale (wind) | -0.177130 | 0.053028 | -3.340 | 0.000983 *** |
| scale(Tair) | 0.009968 | 0.088540 | 0.113 | 0.910466 |

**Case 2:** *lm(formula = scale(NEE) ~ scale(CO2) + scale(fdifRad) + scale(wind))*

| Coefficients: | Estimate | Std. Error | t-value | Pr (>|t|) |
|---|---|---|---|---|
| (Intercept) | -0.10971 | 0.04848 | -2.263 | 0.024591 * |
| scale(CO2) | 0.17776 | 0.05231 | 3.399 | 0.000803 *** |
| scale(fdifRad) | 0.11620 | 0.04873 | 2.385 | 0.017930 * |
| scale (wind) | -0.17344 | 0.04763 | -3.641 | 0.000338 *** |

**Case 3:** *lm(formula = NEE ~ VPD + CO2 + fdifRad + wind + Tair - 1)*

| Coefficients: | Estimate | Std. Error | t-value | Pr (>|t|) |
|---|---|---|---|---|
| VPD | 0.126540 | 0.057204 | 2.212 | 0.02798 * |
| CO2 | 0.014808 | 1.753 | 1.753 | 0.08095 |
| fdifRad | 2.144689 | 1.297013 | 1.654 | 0.09964 |
| wind | -1.635912 | 0.288365 | -5.673 | 4.37e-08 *** |
| Tair | -0.313711 | 0.114494 | -2.740 | 0.00665 ** |

*Case 4: lm(formula = NEE ~ VPD + CO2 + fdifRad + wind + Tair)*

| Coefficients: | Estimate | Std. Error | t-value | Pr (>|t|) |
|---|---|---|---|---|
| (*Intercept*) | -19.78924 | 6.57646 | -3.009 | 0.002926 ** |
| *VPD* | 0.01612 | 0.06711 | 0.240 | 0.810408 |
| *CO2* | 0.03960 | 0.01169 | 3.387 | 0.000837 *** |
| *fdifRad* | 2.99722 | 1.30513 | 2.296 | 0.022589 * |
| *wind* | -1.11112 | 0.33264 | -3.340 | 0.000983 *** |
| *Tair* | 0.01772 | 0.15742 | 0.113 | 0.910466 |

*Case 5: lm(formula = NEE ~ CO2 + fdifRad + wind)*

| Coefficients: | Estimate | Std. Error | t-value | Pr (>|t|) |
|---|---|---|---|---|
| (*Intercept*) | -19.11845 | 4.67333 | -4.091 | 6.01e-05 *** |
| *CO2* | 0.03958 | 0.01165 | 3.399 | 0.000803 *** |
| *fdifRad* | 2.86305 | 1.20054 | 2.385 | 0.017930 * |
| *wind* | -1.08794 | 0.29880 | -3.641 | 0.000338 *** |

| Case number | Goodness of fit | Insignificant p-values | AIC score |
|---|---|---|---|
| *1* | 0.20 | Temperature, *VPD* [0.8 to 0.9] | 494 |
| *2* | 021 | none | 490 |
| *3* | 0.74 | none | 808 |
| *4* | 0.20 | Temperature, *VPD* [0.8 to 0.9] | 801 |
| *5* | 0.21 | none | 798 |

AIC scores differed substantially between models that used original and scaled data, where the model that used the scaled data had low values of *AIC* score. The model (case 3) that used the original data but excluded the intercept had a relatively high value of goodness of fit when compared with all other cases. Because the *AIC* score didn't change much between cases 3 and 4 and that case 3 had a relatively high goodness of fit value, we chose to use the model in case 3 for this study.

[revised manuscript text omitted]